# Breaking the Reclustering Barrier in Centroid-based Deep Clustering

**Lukas Miklautz**[†,1] **Timo Klein**[†,1,2] **Kevin Sidak**[†,1,2] **Collin Leiber**[3,4]
**Thomas Lang**[1,2] **Andrii Shkabrii**[1] **Sebastian Tschiatschek**[‡,1,5] **Claudia Plant**[‡,1,5]

[1] Faculty of Computer Science, University of Vienna, Vienna, Austria
[2] UniVie Doctoral School Computer Science, University of Vienna, Vienna, Austria
[3] Department of Computer Science, Aalto University, Espoo, Finland
[4] Department of Computer Science, University of Helsinki, Helsinki, Finland
[5] ds:UniVie, Vienna, Austria
[†] Joint first authors, [‡] Joint last authors
`firstname.lastname@univie.ac.at, collin.leiber@aalto.fi`

## Abstract

This work investigates an important phenomenon in centroid-based deep clustering (DC) algorithms: Performance quickly saturates after a period of rapid early gains. Practitioners commonly address early saturation with periodic reclustering, which we demonstrate to be insufficient to address performance plateaus. We call this phenomenon the "*reclustering barrier*" and empirically show *when* the reclustering barrier occurs, *what* its underlying mechanisms are, and *how* it is possible to **B**reak the **R**eclustering **B**arrier with our algorithm BRB. BRB avoids early over-commitment to initial clusterings and enables continuous adaptation to reinitialized clustering targets while remaining conceptually simple. Applying our algorithm to widely-used centroid-based DC algorithms, we show that (1) BRB consistently improves performance across a wide range of clustering benchmarks, (2) BRB enables training from scratch, and (3) BRB performs competitively against state-of-the-art DC algorithms when combined with a contrastive loss. **We release our code and pre-trained models at** `https://github.com/Probabilistic-and-Interactive-ML/breaking-the-reclustering-barrier`.

## 1 Introduction

"To live is to change; to be perfect is to change often" (Newman, 1845): Though not originally about ML, this proverb underscores the importance of adaptability. This adaptability is critical for the success of machine learning algorithms, especially during training. It is particularly crucial in clustering — a family of unsupervised learning algorithms that partition samples into multiple groups based on their similarity. Deep clustering (DC) involves not only assigning samples to a particular group but also jointly learning a representation of the data using deep learning. As this representation improves, the algorithm enhances its ability to identify clusters in the data, requiring it to adjust its assignments frequently.

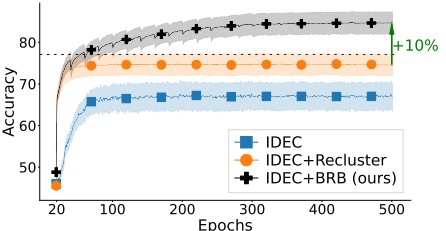

Figure 1: **Breaking the reclustering barrier with BRB.** Uniting the deep clustering method IDEC with BRB (IDEC+BRB) breaks through the performance barrier (dashed line) encountered when just using IDEC or combining it with (IDEC+Recluster). Performance is measured by clustering accuracy on USPS.

In this paper, we analyze whether centroid-based DC algorithms can sufficiently adapt during training to facilitate further improvements. The central finding of our analysis is that they cannot and that commonly used techniques, such as periodically reclustering all samples in the embedded space with $k$-Means, are insufficient to address this issue. Specifically, we find that reclustering by itself

is not enough to enable late-training improvements to the clustering because it fails to change the structure of the underlying embedded space. A bad initial representation or clustering exacerbates the effect and potentially decreases final performance by more than 10%, as our experiments demonstrate. When a clustering algorithm cannot improve late in training despite frequent reclustering, we refer to this as the *reclustering barrier*.

But can we overcome the reclustering barrier? Yes, as we show in Figure 1 for the centroid-based DC method IDEC with reclustering, without reclustering, and with our proposed approach. Our novel method, BRB, can **B**reak the **R**eclustering **B**arrier (BRB). After 100 epochs, BRB breaks through the plateau that cannot be surpassed by the competitors even after 400 additional epochs of training. By analyzing the evolution of the latent space during clustering, we identify a lack of change in the embedding as the primary cause of reclustering's limited effectiveness. BRB combines reclustering with a carefully designed weight reset, leveraging a synergistic effect between these algorithmic components. This synergy creates a virtuous cycle between generating new clustering targets and adapting to them, enabling our method to escape suboptimal performance plateaus. At the same time, BRB is easy to implement and compatible with a wide range of centroid-based DC algorithms.

Our experiments show that despite its simplicity, BRB improves the performance of the widely-used centroid-based DC algorithms DEC (Xie et al., 2016), IDEC (Guo et al., 2017), and DCN (Yang et al., 2017) across a diverse set of benchmarks. When combined with contrastive learning and self-labeling (Gansbeke et al., 2020), BRB even pushes DEC, IDEC, and DCN to state-of-the-art performance. With BRB, these algorithms find good clustering solutions *even without the usual pre-training*, effectively escaping the reclustering barrier. To summarize our contributions:

- We propose BRB, a novel algorithm that can break through existing performance plateaus in centroid-based deep clustering.
- We empirically identify the underlying causes of the reclustering barrier and explain the success of BRB: It preserves the variation within clusters in early training and increases exploration of diverse clustering solutions.
- We apply BRB on top of several deep clustering algorithms, yielding robust performance improvements across a wide range of datasets and setups. With contrastive learning and self-labeling, BRB achieves competitive performance compared to state-of-the-art approaches.

**Research focus** Centroid-based deep clustering is a highly active research area intersecting with various fields. For instance, the current state-of-the-art image clustering technique SeCu (Qian, 2023) is centroid-based. Self-supervised learning uses prototypes, like centroids, for representation learning (Asano et al., 2019; Caron et al., 2018; 2020; 2021). This study concentrates on established centroid-based DC algorithms, specifically DEC, IDEC, and DCN, which are widely used in diverse domains such as biology (Li et al., 2020; Hu et al., 2021; Luo et al., 2021), medicine (Wachowiak et al., 2019; Kalweit et al., 2023) or finance (Choi & Renelle, 2019). These algorithms also form the foundation for several recent DC methods (Qian, 2023; Li et al., 2019). Enhancing DEC, IDEC, and DCN could significantly impact multiple research fields and accelerate progress on downstream applications. With this context, we proceed to discuss some background and related work.

## 2 BACKGROUND AND RELATED WORK

Deep clustering (DC) is the combination of clustering and deep learning. Popular deep learning approaches for DC include self-supervised networks such as autoencoders (AEs) or SimCLR (Chen et al., 2020) as they can be trained without given labels. The idea of centroid-based DC is to obtain an initial clustering, typically by running the clustering algorithm $k$-Means in the latent space of a pre-trained network. The algorithm proceeds to update this result by optimizing neural network parameters $\theta$ and centroids $\mathbf{M}$ through a loss of the form $L(\theta, \mathbf{M}) = \lambda_1 L_{SSL}(\theta) + \lambda_2 L_C(\theta, \mathbf{M})$, where $L_C$ refers to the clustering loss and $L_{SSL}$ refers to the self-supervised loss (e.g., contrastive or reconstruction loss, for SimCLR and AE respectively). Here, the embedding and the clustering can be updated simultaneously or iteratively (Zhou et al., 2022). A well-known representative utilizing simultaneous optimization is DEC (Xie et al., 2016). It uses a kernel based on the Student's t-distribution to minimize the Kullback-Leibler divergence between the data distribution and an auxiliary target distribution, which can be trained end-to-end. As $L_{SSL}$ is not used during the clustering optimization of DEC, a distorted embedding could occur (Guo et al., 2017). This issue is

tackled by IDEC (Guo et al., 2017), which concurrently optimizes $L_C$ and $L_{SSL}$. DCN (Yang et al., 2017) pursues an iterative optimization, where the embedding is frozen when the clustering is updated, and vice versa. Since the cluster centers are not learned via the neural network but explicitly updated, it is feasible to use $k$-Means to obtain non-differentiable hard cluster labels. The well-established algorithms DEC, IDEC, and DCN are the building blocks for a number of follow-up works and have diverse applications, making them a natural choice for this study. We provide more details on these algorithms in Appendix B.1.

Contemporary DC methods broadly fall into two groups. The first group is focused on image clustering (Gansbeke et al., 2020; Dang et al., 2021; Li et al., 2021; Zhong et al., 2021; Qian, 2023) and leverages domain knowledge, such as augmentations. The second group consists of data-agnostic DC methods, like DEC, IDEC, and DCN or (Miklautz et al., 2021; Leiber et al., 2021; Mahon & Lukasiewicz, 2024) that do not rely on augmentations. For a more detailed overview of DC algorithms, see (Ren et al., 2022; Zhou et al., 2022; Lu et al., 2024).

## 3 PROBLEM SETUP

Given an unlabeled dataset $\mathcal{X} = \{\mathbf{x}_i\}_{i=1}^n$ containing $n$ instances $\mathbf{x}_i \in \mathbb{R}^D$, our objective is to partition $\mathcal{X}$ into $k$ distinct clusters. Each cluster is represented by a centroid $\boldsymbol{\mu}_j \in \mathbb{R}^d$, with $j \in \{1, \ldots, k\}$. This partitioning requires learning assignments $s_{i,j} \in \{0, 1\}$ of instances $\mathbf{x}_i$ to centroids $\boldsymbol{\mu}_j$ subject to $\sum_{j=1}^k s_{i,j} = 1$ for all $i$. Additionally, *deep* clustering methods aspire to learn a "cluster-friendly" (Yang et al., 2017) latent feature space $\mathcal{H} \subseteq \mathbb{R}^d$, where $d \ll D$ utilizing a non-linear encoder $f_\theta \colon \mathcal{X} \to \mathcal{H}$. For example, a $k$-Means friendly latent space consists of compressed spherical clusters (low intra-cluster distance) that are well-separated (high inter-cluster distance).

Many methods utilize a self-supervised auxiliary objective to prevent trivial solutions in DC, e.g., by including a reconstruction or contrastive loss in addition to the clustering loss. This auxiliary objective is applied in a task-dependent output space $\mathcal{Z}$ and is often learned with a separate task head $g \colon \mathcal{H} \to \mathcal{Z}$. In the case of reconstruction, the function $g$ serves as a decoder network, where $\mathcal{Z} \subseteq \mathbb{R}^D$ corresponds to the space of reconstructed data points. For contrastive learning, $g$ acts as a projector network (e.g., a shallow MLP), where $\mathcal{Z}$ is the projector's output space.

A deep centroid-based clustering algorithm as outlined above can be formalized as the iterative application of a function $\mathcal{C} \colon (\theta_t, \mathbf{S}_t, \mathbf{M}_t) \mapsto (\theta_{t+1}, \mathbf{S}_{t+1}, \mathbf{M}_{t+1})$. Here, $\theta_t \in \Theta$ represents the network parameters at step $t$, $\mathbf{M}_t \in \mathbb{R}^{k \times d}$ the centroid matrix and $\mathbf{S}_t \in [0, 1]^{n \times k}$ a matrix encoding the cluster assignments for each sample. The function $\mathcal{C}$ optimizes a clustering objective $L(\theta_t, \mathbf{S}_t, \mathbf{M}_t) \colon \Theta \times [0, 1]^{n \times k} \times \mathbb{R}^{k \times d} \to \mathbb{R}$, utilizing a gradient-descent optimizer. Its output is improved parameters $\theta_{t+1}$ as well as new assignments $\mathbf{S}_{t+1}$ and centroids $\mathbf{M}_{t+1}$. Typically, $\mathcal{C}$ is iterated until a local minimum of $L(\theta, \mathbf{S}, \mathbf{M})$ is reached.

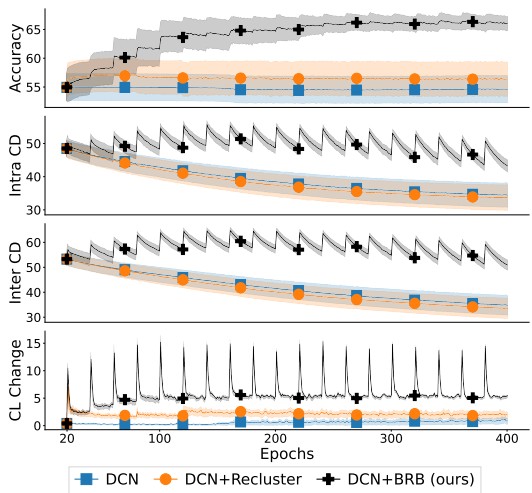

Figure 2: **Why does reclustering not work?** DCN with reclustering (orange line) shows minimal changes compared to unmodified DCN (blue line) in the embedded space (intra/inter-CD) or explored clusterings (CL Change) for GTSRB. This effect generalizes to other datasets (Figure 6).

## 4 THE RECLUSTERING BARRIER

Before introducing BRB in Section 5, we examine why reassigning all samples in the embedded space via, e.g., $k$-Means by itself fails to enhance performance beyond a certain threshold during DC training. We refer to the aforementioned procedure as "*reclustering*" and assess its effect on the embedded space using intra-class distance (intra-CD) to measure variation within ground truth classes and inter-class distance (inter-CD) to assess separation between ground truth

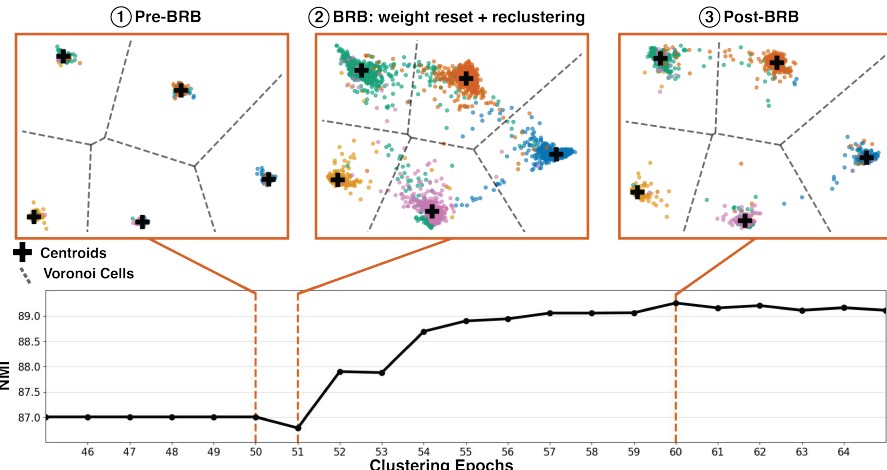

Figure 3: **The effect of BRB during training of DEC.** *(1) Pre-BRB*: Before BRB is applied, DEC strongly compresses (low variation within clusters) the five shown clusters and mixes different ground truth classes of USPS (indicated by color), leading to a performance plateau (cf. epochs 46 to 50 in the NMI plot). *(2) BRB* applies weight reset to increase the variation in the clusters with subsequent reclustering, which leads, after a small performance drop in NMI (epoch 51), to a steep increase (epochs 52 to 59). *(3) Post-BRB*: After applying BRB, the clusters are compressed again by DEC until BRB is applied another time.

classes; this follows the embedding space analysis in (Lehner et al., 2024). We quantify the impact of reclustering on the clustering targets by the *cluster label change* (CL Change) between epochs in terms of the normalized mutual information (NMI) (Kvalseth, 1987) as $(1 - \mathrm{NMI}(\mathbf{S}_t, \mathbf{S}_{t-1})) \cdot 100$. A value of 0 means no change between two consecutive clusterings, while a value of 100 indicates complete change. Figure 2 tracks these three metrics and the clustering accuracy (Yang et al., 2010) with BRB application every $T = 20$ epochs on the GTSRB dataset [1].

Although reclustering marginally enhances clustering accuracy (orange line), it does not substantially alter the embedded space: the intra-CD and inter-CD are almost the same at the end of the training, irrespective of whether reclustering is applied. Correspondingly, the CL Change is largely unaffected by the reclustering, indicating only marginal differences in the obtained clustering solutions. These findings suggest that if we want to further improve performance, we require larger and better-structured changes in the embedded space to systematically impact the clustering. In contrast to only reclustering, BRB induces such changes and is able to improve the clustering performance (black line in Figure 2). Figure 3 schematically illustrates the changes caused by BRB in the embedded space. In Section 5, we describe how our approach, BRB, effectively introduces such changes.

> **Reclustering Barrier**
>
> Strongly compressed and well-separated clusters in the embedded space prevent reclustering from discovering new solutions late in training. Figure 3 provides an intuition.

## 5 OUR APPROACH - BRB

Our goal with this work is to overcome the reclustering barrier in a centroid-based deep clustering algorithm $\mathcal{C}$. To this end, we establish three desiderata for our approach:

1. *Exploration*: The proposed modification should increase the exploration of clustering solutions.
2. *Knowledge preservation*: The knowledge encoded in network parameters $\theta$ should be preserved while exploring a wider range of solutions.
3. *Generality*: The proposed modification should be applicable to existing DC approaches.

---

[1] Section 6.1 describes all datasets used in this work in detail.

We now define our modified algorithm as

$$\mathcal{C}_{\mathrm{BRB}} : (\theta_t, \mathbf{S}_t, \mathbf{M}_t) \mapsto \mathcal{C}(\iota_w(\theta_t), \iota_c(\mathbf{S}_t, \mathbf{M}_t)),$$

consisting of two modifications $\iota_w$ and $\iota_c$ as well as optional momentum resets, each impacting different components of the deep clustering algorithms.

**Weight resets**  Section 4 motivates the need for structured changes in the embedded space to enable effective reclustering. BRB achieves such changes by applying a soft weight reset (Ash & Adams, 2020; D'Oro et al., 2023) to the the non-linear encoder $f_{\theta_t}$. In particular, we obtain modified parameters $\tilde{\theta}_t^i$ through a convex combination of the network's prior parameters and freshly sampled weights (D'Oro et al., 2023), denoted as $\iota_w(\theta_t^i)$:

$$\tilde{\theta}_t^i = \iota_w(\theta_t^i) = \alpha\theta_t^i + (1-\alpha)\phi^i, \tag{1}$$

where $\theta_t^i$ are layer $i$'s weights at step $t$, $\phi^i$ is sampled from the initial weight distribution and $\alpha \in (0,1)$ is a hyperparameter. This strategy balances the need for inducing changes in the embedding function $f_{\theta_t}$ to escape local minima while preserving previously learned information.

While Equation 1 outlines a particular reset strategy, our approach is capable of accommodating different reset techniques, such as adding noise (cf. Section 6.3) or layer-wise resets (Alabdulmohsin et al., 2021). We discuss our choices in the paragraph "Implementation details" later in this section.

**Reclustering**  Applying a weight-reset $\iota_w(\theta_t)$ to the encoding function $f_{\theta_t}$ induces a shift in embeddings: $f_{\tilde{\theta}_t}(\mathbf{X}) = \tilde{\mathbf{H}}_t \neq \mathbf{H}_t = f_{\theta_t}(\mathbf{X})$. Consequently, the centroids $\mathbf{M}_t$ at clustering step $t$ do not accurately characterize the cluster centers for $\tilde{\mathbf{H}}_t$. We address this issue by proposing to recalculate the centroids based on the updated embedding matrix $\tilde{\mathbf{H}}_t$:

$$\tilde{\mathbf{S}}_t, \tilde{\mathbf{M}}_t = \iota_c(\mathbf{S}_t, \mathbf{M}_t, \tilde{\theta}_t) = \mathrm{recluster}\left(f_{\tilde{\theta}_t}(\mathbf{X})\right) \tag{2}$$

Here, $\tilde{\mathbf{M}}_t$ denotes the centroid matrix obtained from reclustering with the perturbed embeddings $f_{\tilde{\theta}_t}(\mathbf{X}) = \tilde{\mathbf{H}}_t$ and $\tilde{\mathbf{S}}_t \in [0,1]^{n \times k}$ represents the corresponding assignments. Any centroid-based method, such as $k$-Means (Lloyd, 1982) or $k$-Medoids (Rdusseeun & Kaufman, 1987), is applicable for reclustering. However, we choose $k$-Means in BRB for two key reasons. First and foremost, it aligns with the spherical cluster model and initialization used by DEC, IDEC, and DCN, e.g., DCN already uses $k$-Means in its assignment step. Second, our ablation study (Appendix H.14) shows that $k$-Means leads to better exploration of clustering solutions and improved performance compared to the assignment procedures of DEC and IDEC.

**Momentum resets**  DEC and IDEC parameterize and iteratively refine cluster centers through gradient descent. However, integrating momentum-based optimization with weight resets and reclustering may introduce misalignment between momentum terms and re-calculated centroids, for which we provide ablations in Appendix H.2. To mitigate this, we re-initialize the momentum terms for the cluster centers to zero after reclustering. This aligns gradient steps and recalculated centroids, preventing the moment estimates from diverging and harming task performance (Lyle et al., 2023). For DCN, which does not parameterize centroids, momentum resets are unnecessary.

**Implementation details**  The pseudo-code in Algorithm 1 shows how BRB is added to the training loop of an exemplary DC algorithm, highlighting its straightforward implementation and minimal overhead. We use four key implementation details for BRB that we highlight in the following.

(1) Resetting the encoder's final embedding layer can adversely affect performance due to excessive perturbations in the latent space (see Appendix H.1). We observe a similar phenomenon in deep CNN architectures like ResNet (He et al., 2016), where resetting all layers also induces excessive perturbations (see Appendix H.3). Our solution is to only reset the last ResNet Block and the MLP encoder. We (re-)initialize all weights using Kaiming uniform initialization (He et al., 2015), which is Pytorch's default (Paszke et al., 2019).

(2) We introduce a reset interval hyperparameter $T$ and reset only every $T$-th epoch. For the reclustering step, we use the $k$-Means algorithm (Lloyd, 1982) due to its simplicity,

performance, and speed. Appendix Table 10 shows a runtime comparison of different reclustering algorithms, while Appendix Table 12 shows a performance comparison.

(3) We sub-sample the dataset when reclustering to enhance computational efficiency. We use a subsample size of 10,000 for all datasets, which leads to a negligible runtime overhead of approximately 1.1% when using BRB. In Tables 4 and 9 in Appendix G, we analyze the impact of the subsample size on clustering accuracy and runtime performance in detail.

(4) Using image augmentation is a common enhancement for DC algorithms, and we find it complements BRB as well. We use augmentations for all baselines and BRB by augmenting each sample and enforcing consistent cluster assignments between a sample and its augmented counterparts (see Appendix E for details). We split datasets into grayscale and color (cf. Section 6.1), applying a random affine transformation for grayscale data and the SimCLR CIFAR augmentations (Chen et al., 2020) for color data. Exact augmentation parameters are described in Tables 5,6, and 7, respectively.

# 6  EXPERIMENTS

We integrate BRB into DEC, IDEC, and DCN, presenting experiments that demonstrate the performance improvements resulting from this integration for a wide range of datasets. The first part of our analysis describes the datasets and setups used in our experiments. We then proceed to benchmark BRB against unmodified versions of DEC, IDEC, and DCN in three scenarios: With and without pre-training and with a contrastive auxiliary task. Then, we dig deeper into the underlying mechanisms of BRB and examine how it is able to break the reclustering barrier. **Overall, we find that BRB improves performance in 88.10 % of all runs while incurring a minimal runtime overhead of approximately 1.1 %.**[2]

## 6.1  EXPERIMENT SETUP AND DATASETS

We evaluate BRB on eight common deep clustering datasets, divided into two groups, each with a distinct setup. Dataset and preprocessing details are provided in Appendix C. The first group (MNIST (LeCun et al., 1998), KMNIST (Clanuwat et al., 2018), FMNIST (Xiao et al., 2017), USPS (Hull, 1994), OPTDIGITS (Alpaydin & Kaynak, 1998), and GTSRB (Stallkamp et al., 2012)) uses a feed-forward autoencoder with reconstruction as an auxiliary task, a learning rate of 0.001, and **fixed BRB hyperparameters** $\alpha = 0.8$ **and** $T = 20$ **across all algorithms and setups**. The second group (CIFAR10 and CIFAR100-20 (Krizhevsky et al., 2009)), consisting of more challenging color image datasets, uses SimCLR (Chen et al., 2020) as an auxiliary task, a ResNet18 (He et al., 2016) encoder, and self-labeling (Gansbeke et al., 2020) after clustering as in (Qian, 2023). See Appendix E for full experimental details.

Given the datasets outlined above, we evaluate DEC, IDEC, and DCN with and without BRB in three scenarios: (1) starting from a representation pre-trained with reconstruction as an auxiliary task, (2) training from scratch with reconstruction as auxiliary task, and (3) starting from a representation pre-trained with contrastive learning. Scenario (1) examines whether BRB can break through performance plateaus late in training after pre-training took place. Scenario (2) tests whether BRB is able to improve the exploration of clustering solutions. When training from scratch, we expect the baselines to perform poorly due to premature convergence to suboptimal solutions. BRB, however, should leverage resets and reclustering to continuously escape local minima and achieve better performance. In scenario (3), we use contrastive pre-training to scale our approach to more challenging datasets.

We measure algorithm performance using three standard clustering metrics (Lu et al., 2024; Huang et al., 2024; Zhou et al., 2025):accuracy, Adjusted Rand Index (ARI) (Hubert & Arabie, 1985), and Normalized Mutual Information (NMI) (Kvalseth, 1987). All metrics are scaled between $[0, 100]$, where higher values indicate better performance. Results are reported as the average over ten runs, with standard errors unless otherwise noted. Similar to Lehner et al. (2024), we track the separation of ground truth clusters in the embedded space with the inter/intra-class distances (inter-CD/intra-CD), which measure the variation between and within classes, respectively. We measure the change in

---

[2]Appendix G provides a detailed theoretical and empirical runtime analysis.

cluster labels (CL Change) during training using the NMI of cluster labels in subsequent epochs. See Section 4 for details on inter/intra-CD and cluster label change, and Appendix D for a full description of all metrics.

## 6.2 BENCHMARK RESULTS

**Scenario 1: with pre-training** Figure 4 compares the performance of BRB against baselines on several DC benchmark datasets. As BRB consists of a weight reset and subsequent reclustering, we provide ablation studies for each component individually. Our results suggest that neither reclustering nor resetting can consistently improve the baseline on their own. For example, on GTSRB, *DCN+Reset* performs much worse than the baseline (-10%), while *DCN+Recluster* is surpassed by the baseline on USPS.

In contrast, BRB leverages the synergy between weight resets and reclustering and improves performance across all datasets. The ablation experiments for IDEC and DEC confirm the average improvement of BRB over the unmodified baseline and are shown in Appendix Table 13.

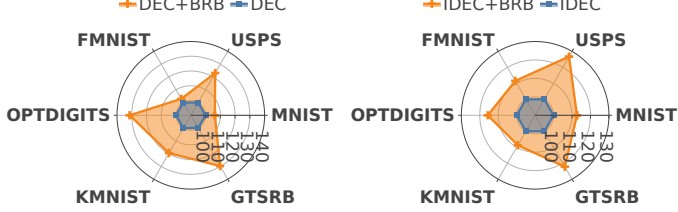

**Scenario 2: without pre-training** DEC, IDEC, and DCN heavily rely on pre-training to get sufficiently good initial clusters. To our surprise, we find that BRB reaches strong levels of performance even without pre-training. Figure 5 shows the relative improvement in clustering accuracy for DEC and IDEC when compared to the unmodified algorithms, with absolute values shown in Table 13. Without BRB, the perfor-

Figure 4: **Improved performance of BRB.** Relative change in average accuracy against the unmodified DCN algorithm for *BRB* and important ablations **with pre-training**: *DCN+Reset* refers to performing only weight resets (Eq. 1); *DCN+Recluster*, refers to only reclustering. Our *DCN+BRB* with both weight resets and reclustering consistently improves performance for DCN.

mance of DEC and IDEC plateaus early without pre-training, as highlighted already in Figure 1 for IDEC. BRB significantly improves the performance for both algorithms, sometimes by up to 30%, e.g., when using DEC on OPTDIGITS or GTSRB and for IDEC on USPS. Table 13 underscores that these large improvements hold for DCN as well. Interestingly, BRB improves DEC's accuracy from 61 to 77 on OPTDIGITS without image augmentation, showing it can build strong representations from random initialization. However, we still recommend using image augmentations for generally better results.

**Scenario 3: with contrastive learning** All of the above experiments use an AE with reconstruction loss as an auxiliary task. Table 1 shows that BRB's performance improvements also transfer to other auxiliary tasks. Here, we compare DEC, IDEC, and DCN with and without BRB using SimCLR and self-labeling

Figure 5: **BRB enables training from scratch.** Relative improvements of clustering accuracy for DEC and IDEC when using BRB **without pre-training**.

(Gansbeke et al., 2020); see Section E for implementation details and Table 7 for specific hyperparameter settings. We follow the experiment setup of Qian (2023) by using a ResNet18 for all contrastive experiments and reporting the best performance after self-labeling on the respective test sets of CIFAR10 and CIFAR100-20. In Table 1, we see that BRB consistently improves performance for IDEC and DCN by about 2-3%, while DEC benefits from BRB for CIFAR10 by more than 2%. Surprisingly, we find that the established methods IDEC and DCN outperform more recent deep clustering algorithms when combined with a contrastive task and BRB. To the best of our knowledge, this is the first time that such high performance has been reported for DEC, IDEC, and DCN on CIFAR10 and CIFAR100-20 using a ResNet18. The findings highlight the continued relevance of these "older" methods up to today. For CIFAR100-20, **DCN+BRB and IDEC+BRB even beat the current state-of-the-art** deep clustering method SeCu (Qian, 2023). We provide additional results in Appendix Table 16 and results without self-labeling in Appendix Table 17.

Table 1: **Clustering performance with contrastive task and self-labeling**. BRB improves overall performance for most baselines and DCN+BRB outperforms the state-of-the-art method SeCu on CIFAR100-20. The per-method best results are in bold, and the overall best results are underlined.

| Methods | CIFAR10 | | | CIFAR100-20 | | |
|---|---|---|---|---|---|---|
| | ACC | NMI | ARI | ACC | NMI | ARI |
| Pretraining + $k$-Means | 68.97 | 63.98 | 40.13 | 37.22 | 42.25 | 14.86 |
| DEC | 88.29 | 80.60 | 77.23 | 50.16 | 51.66 | **35.37** |
| DEC + BRB | **90.57** | **82.57** | **81.18** | **50.46** | **51.72** | 35.05 |
| IDEC | 88.30 | 79.50 | 77.27 | 52.73 | 52.79 | 36.79 |
| IDEC + BRB | **90.72** | **83.26** | **81.81** | **55.43** | **54.81** | **38.81** |
| DCN | 88.55 | 81.02 | 78.17 | 53.27 | 52.13 | 37.30 |
| DCN + BRB | **91.23** | **83.66** | **82.42** | **56.92** | **56.76** | **41.15** |
| SCAN (Gansbeke et al., 2020) | 88.3 | 79.7 | 77.2 | 50.7 | 48.6 | 33.3 |
| GCC (Zhong et al., 2021) | 90.1 | - | - | 52.3 | - | - |
| SeCu (Qian, 2023) | 93.0 | 86.1 | 85.7 | 55.2 | 55.1 | 39.7 |

**Additional experimental results** BRB introduces two key hyperparameters: reset strength $\alpha$ and reset interval $T$. These parameters require a trade-off between sufficient perturbation of the embedded space and algorithm recovery. Our hyperparameter sensitivity analysis in Appendix E.2 demonstrates BRB's stability for moderate reset strengths ($0.8 \leq \alpha \leq 0.9$) and intervals ($T \in \{10, 20, 40\}$). We use these ranges for all experiments, with both autoencoders and contrastive learning. In Appendix G, we validate that BRB is lightweight and adds only minor computational overhead by performing a detailed **runtime analysis**. Lastly, we also study the impact of different reclustering algorithms in Table 10.

## 6.3 ANALYSIS OF BRB

Our central hypothesis is that BRB's soft resets and subsequent reclustering structurally alter the embedded space and enable finding new clustering solutions. We validate this hypothesis by analyzing changes in the embedded space via the inter-class distance (inter-CD), with *higher* values indicating enhanced separation, and measuring variation within ground truth classes with the intra-class distance (intra-CD). BRB's impact on clustering assignments is measured using the NMI-based cluster label change (CL Change) metric, detailed in Section 4. As BRB's impact differs slightly depending on the dataset, we choose to analyze datasets where it has a substantial, moderate, and minor impact on performance. Therefore, we focus our analysis of BRB on the DCN algorithm and GT-SRB (Figure 2), OPTDIGITS, and MNIST datasets (Figure 6).

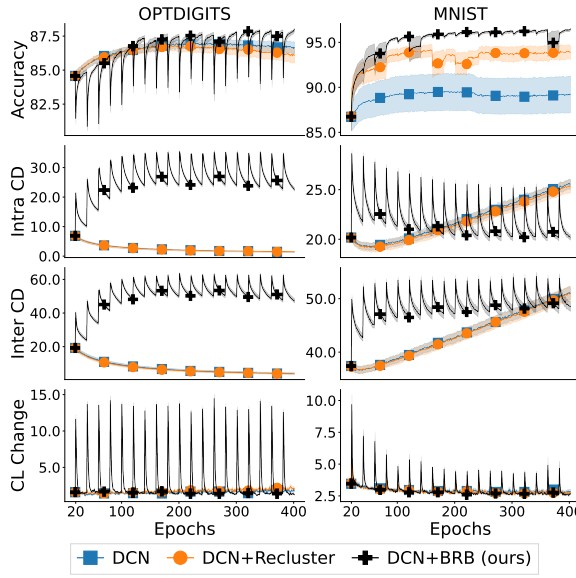

Figure 6: **Effects of BRB.** We track average clustering accuracy, inter/intra-class distance (inter/intra-CD), and cluster label change (CL Change).

**Early over-commitment** For all three considered datasets, BRB increases intra-CD after the first few resets, counteracting premature cluster compression and potential early misassignments. For GTSRB and OPTDIGITS, the intra-CD increases over the whole training, whereas for MNIST, the intra-CD decreases around epoch 60. This leads to a lower intra-CD at the end of the training compared to the baselines. We find that avoiding cluster compression is more important in early training than later on, judging from BRB outperforming the baselines on MNIST despite a lower intra-CD at the end of training. We analyze the specific behavior of the increasing intra-CD and inter-CD for DCN and DCN+Recluster on MNIST in more detail in Appendix H.10.

One possible hypothesis as to why BRB is effective at preventing early over-commitment is that its resets move samples closer to other centroids, allowing for more frequent reassignments during training. To verify this hypothesis, we look at the ratio $\rho(\mathbf{x}_i) = d_1(\mathbf{x}_i)/d_2(\mathbf{x}_i)$ between the distance to the closest centroid $d_1(\mathbf{x}_i)$ and the second closest centroid $d_2(\mathbf{x}_i)$ for a sample $\mathbf{x}_i$. If this ratio is close to 1, then $\mathbf{x}_i$ lies directly between two centroids and has a high likelihood of potentially being reassigned. If $\rho(\mathbf{x}_i) = 0$, then the sample's position in the embedded space is directly on top of the centroid that it is assigned to. Figure 7 shows how $\rho$ is distributed before and after applying BRB during DCN training. We choose DCN for this experiment because its clustering notion relies on the Euclidean distance, which is easy to interpret compared to the

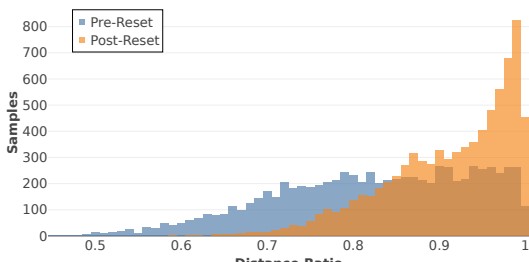

Figure 7: **Effects of BRB on centroid distances.** Distribution of distance ratios $\rho(\mathbf{x}_i) = d_1(\mathbf{x}_i)/d_2(\mathbf{x}_i)$ for the distance to the closest $d_1(\mathbf{x}_i)$ and second-closest centroid $d_2(\mathbf{x}_i)$. BRB prevents early over-commitment by distributing samples more evenly between centroids, allowing for easier reassignment.

Student-t kernel of DEC or IDEC. Looking at the distribution of distance ratios $\rho$ in Figure 7, we see that after applying BRB, there are substantially more samples with a ratio $\rho$ between 0.9 and 1. This means that samples are more evenly distributed between centroids, which in turn facilitates potential reassignments in future update steps, effectively preventing early over-commitment.

**Cluster separation**   Because BRB increases variation within clusters, it needs to separate the classes by a higher margin in order to achieve the same or a better clustering accuracy with less compressed clusters (higher intra-CD). Consequently, Figures 2 and 6 indicate that BRB leads to higher distances between ground truth clusters as indicated by a higher inter-CD in the third row of both figures. For GTSRB and OPTDIGITS, the high intra-CD leads to a correspondingly higher inter-CD. For MNIST, the training ends with similar inter-CD values as the baselines, whereas the intra-CD at the end of training is lower than the corresponding baseline values.

**Exploration of solutions**   For all three datasets, we observe a significant spike in CL Change after applying BRB. As noted in Section 4, the *Recluster* step minimally changes the clustering compared to the unmodified baseline. We attribute this to reclustering only reinforcing existing solutions rather than generating substantially different ones, thus only slightly increasing CL Change and clustering accuracy. In contrast to the reclustering baseline, BRB manages to explore more diverse clusterings (higher CL Change) during training while retaining the knowledge of previous clustering solutions.

## 6.4   ABLATION STUDIES

We conduct four ablation studies to investigate the impact of different components on our method. First, we isolate the role of new cluster labels in shaping the cluster structure while keeping the encoder fixed (Appendix F). Second, we replace soft resets with non-persistent noise injections (Appendix F). Third, we ablate the individual components of Equation (1) in Appendix H.13. Finally, we examine the effect of reclustering with $k$-Means for DEC and IDEC( Appendix H.14). Figure 10 shows that both disentangling label changes from weight resets and replacing soft resets with noise leads to performance degradation below 80% accuracy on OPTDIGITS. BRB, on the other hand, maintains its ability to adapt to new targets, achieving over 85% accuracy. We find that disentangled labels suffer from insufficient label changes, while the non-persistent noise introduces excessive changes late in training. Neither modification affects intra-cluster or inter-cluster distances. Figures 19a and 19b show the impact of removing the contraction component, i.e., the left side of Equation (1) and replacing the weight perturbation with scaled Gaussian noise. Removing contraction consistently hinders training, while Gaussian noise degrades performance on USPS. This highlights the positive effect of soft resets on gradient updates, as observed in supervised learning (Ash & Adams, 2020). Furthermore, Appendix H.14 investigates the implications of reclustering with $k$-Means for DEC and IDEC. Unlike DCN, which naturally generates assignments with $k$-Means, DEC and IDEC initially generate soft assignments that are subsequently hardened. Tables 19 and  18 show that BRB with $k$-Means instead of their native soft assignment consistently improves performance.We attribute this to the new $k$-Means labels providing an improved exploration of clustering solutions.

> **BRB Mechanisms**
>
> - BRB prevents early over-commitment by increasing intra-class variance while preserving cluster separation.
> - BRB explores new clustering solutions by inducing cluster label changes without destroying the cluster structure, allowing the reassignment of samples late in training.
> - BRB's resets are necessary for performance improvements. Cluster label changes without corresponding structural changes in the embedding fail to alter the optimization trajectory.

## 7 DISCUSSION

**Broader insights for deep clustering research**   Over-commitment to early clustering targets is of great interest to the deep clustering community, given the use of powerful deep neural networks to fit changing clustering targets. Our study identifies sharp drops in the intra-class distance early in training as the main culprit of this problem. Once objects with different ground truth classes are compressed in a cluster, reclustering is insufficient to separate them again — a phenomenon we term the "*reclustering barrier*". Section 6.4 shows that the network fails to adapt to the more diverse targets even if the variation in clustering labels is increased. While we use these findings to develop BRB, they may help to guide the design of future deep clustering methods as well.

**Connection to early overfitting in other fields**   The *primacy bias* is an example of early overfitting (Nikishin et al., 2022) in deep reinforcement learning. In supervised learning, Zaidi et al. (2022) explore the role of weight resets regarding generalization performance. They find that soft resets enhance test accuracy by preventing overfitting to noise during early training (Zaidi et al., 2022) and hypothesize that resets may force the network to prioritize learning the "easy" data points first. The resets potentially prevent the network from "confidently fitting to [the] noise" (Zaidi et al., 2022). In DC algorithms, fitting to noise corresponds to a premature assignment of points to specific clusters. Subsequent optimization steps, particularly the compression loss $L_C$, inhibit these points from being re-assigned later. BRB mitigates this issue to some extent, as shown by resets increasing the intra-CD.

**Limitations and future work**   BRB is partially inspired by theoretical research on perturbation-robust clustering (Bilu et al., 2013; Awasthi et al., 2012). We hypothesize that BRB's resets and reclustering create an analog to an $\alpha$-perturbation-resilient clustering instance, where the soft resets mimic the $\alpha$-perturbation, altering the Euclidean distance in the embedding space by a factor of at most $\alpha$. A full theoretical analysis of BRB in deep clustering is complex due to the nonlinearity of neural networks and minibatch SGD and would be a significant contribution on its own. Therefore, we focus on empirical results and ablation studies and want to explore the theoretical aspect in future work. Another limitation is that we investigate the *reclustering* barrier for centroid-based DC algorithms only. Whether a similar phenomenon exists for non-centroid-based DC algorithms, such as density-based or agglomerative clustering methods, is an exciting direction for future work.

## 8 CONCLUSION

This paper identifies a common problem in centroid-based DC algorithms: Reclustering during training fails to explore new clustering solutions due to early over-commitment to a sub-optimal clustering. We call this phenomenon the *reclustering barrier*. Integrating our proposed algorithm BRB into widely-established DC methods, like DCN, DEC, and IDEC, demonstrates consistent performance gains under multiple conditions, including a range of different datasets, different auxiliary tasks (contrastive learning and autoencoding), and settings with and without pre-training. Under all of these conditions, BRB is able to break through existing performance plateaus of DEC, IDEC, and DCN. Notably, BRB with contrastive learning yields competitive performance to state-of-the-art algorithms on CIFAR10 and CIFAR100-20.

Through empirical analysis, we identify three key mechanisms contributing to BRB's success. These mechanisms consist of (i) prevention of premature commitment to clustering solutions by DC methods (ii) exploration of a wider range of clusterings and (iii) enabling adaptation to improved clustering targets. BRB can be easily integrated in centroid-based DC methods. Given that DCN, DEC, and IDEC underpin numerous DC algorithms, we anticipate that BRB will impact the development of centroid-based DC methods going forward.

## REPRODUCIBILITY

Our code and models are publicly available on github at `https://github.com/Probabilistic-and-Interactive-ML/breaking-the-reclustering-barrier`. Additionally, we provide pseudocode of our algorithm in Section A and all experiment hyperparameters together with implementation details in Appendix Section E.

## ETHICS STATEMENT

Our work contains foundational research in the field of deep clustering and, as such, has the potential to accelerate scientific advances in many other areas. At the same time, it is difficult to rule out malicious use due to the broad applicability of our work to different use cases. This is especially true as we plan to release pretrained general-purpose models for image feature extraction and clustering. Nevertheless, we believe that the ability of these models to enable future research outweighs the drawbacks of potential malicious use.

Focusing on the medical domain, BRB can improve the grouping of patients into different categories, enabling targeted treatments based on the patient's characteristics. While BRB may improve the capabilities of DC algorithms in this respect, it is imperfect and will still produce errors. In applications such as medicine, where errors are prone to high scrutiny, algorithmic failures may lead to public backlash and slow down machine learning adoption as a whole. Similar concerns arise regarding use cases such as fraud or outlier detection in finance. We therefore recommend human oversight when deploying any machine learning algorithm in the real world. In terms of machine learning research, we believe that our findings are also interesting to other fields where algorithms struggle to deal with early overfitting, such as semi-supervised learning.

### ACKNOWLEDGMENTS

We acknowledge EuroHPC Joint Undertaking for awarding us access to Meluxina at LuxProvide, Luxembourg. Also, we gratefully acknowledge financial support from the Vienna Science and Technology Fund (WWTF-ICT19-041), the Austrian Science Foundation FWF (project I5113), and the Federal Ministry of Education, Science and Research (BMBWF) of Austria within the interdisciplinary project "Digitize! Computational Social Science in the Digital and Social Transformation". We thank the open-source community for providing the tools that enabled this work: PyTorch (Paszke et al., 2019), NumPy (Harris et al., 2020), Matplotlib (Hunter, 2007), and ClustPy (Leiber et al., 2023). Lastly, we want to thank the anonymous reviewers who helped improve our work through their feedback.

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

# Appendix

Table 2 summarizes the contents of our Appendix:

| Appendix Section | Content |
|---|---|
| Appendix A | Pytorch-style pseudocode for BRB |
| Appendix B | Detailed information about the clustering algorithms used in this study
SimCLR (Chen et al., 2020) overview
Overview of resetting methods in other fields |
| Appendix C | Datasets of our experiments, data properties, and preprocessing |
| Appendix D | Evaluation metrics |
| Appendix E | Implementation details
Hyperparameters & Hyperparameter sensitivity analysis |
| Appendix F | Detailed setup and results of the noise & disentanglement ablations |
| Appendix G | Runtime analysis |
| Appendix H | Additional experimental results and figures
Full result tables |

Table 2: Structure of our appendix.

## A PSEUDOCODE

---

**Algorithm 1** PyTorch-style pseudo-code of BRB

---

```
# model: neural network
# dc: deep clustering method
# alpha: BRB reset factor (0 <= alpha <= 1)
# recluster_algorithm: algorithm for reclustering
# T: BRB reset interval
# optimizer: optimizer to be used, default : Adam
# subsample_size: size of sample used for reclustering

# deep clustering training loop
for epoch in epochs:
    if epoch % T == 0 and epoch > 0:
        # perform BRB
        reset_model_weights(model, alpha)
        # embed data with reset model
        emb = embed_data(model, loader, subsample_size)
        # reinitialize cluster centroids
        dc.centroids = recluster_algorithm(emb)
        if dc.centroids.has_momentum():
            # reset momentum of learnable centroids
            reset_momentum(optimizer, dc.centroids)

    # load a minibatch x
    for x in loader:
        # perform deep clustering update steps
        dc.update(x)
```

---

## B BACKGROUND

In this section, we provide additional background for some of the methods employed in our main paper. It is structured as follows: First, we provide more details on deep clustering objective functions in Section B.1. Then, we briefly review SimCLR (Chen et al., 2020) as the contrastive learning method used in this work. Lastly, we review reset methods in other fields.

B.1 DEEP CLUSTERING OPTIMIZATION

In general, the procedures mentioned in this work optimize $L(\theta, \mathbf{M}) = \lambda_1 L_{SSL}(\theta) + \lambda_2 L_C(\theta, \mathbf{M})$. To highlight the differences in more detail, we explain below how $L_C$ is chosen. Applying BRB to DEC, IDEC, or DCN does not change the loss functions defined in Equation (5) and Equation (6) or how backpropagation is performed.

**DEC and IDEC:** As mentioned in Sec. 2, the clustering loss $L_C$ of DEC (Xie et al., 2016) and IDEC (Guo et al., 2017) is based on the Kullback-Leibler divergence. Here, the data distribution $\mathbf{Q}$ is compared with an auxiliary target distribution $\mathbf{P}$. The data distribution $\mathbf{Q}$ is quantified by a kernel based on the Student's t-distribution:

$$q_{i,j} = \frac{(1 + ||\mathbf{z}_i - \boldsymbol{\mu}_j||_2^2)^{-1}}{\sum_{j'}(1 + ||\mathbf{z}_i - \boldsymbol{\mu}_{j'}||_2^2)^{-1}}, \tag{3}$$

where $\mathbf{z}_i$ is an encoded sample, i.e., $\mathbf{z}_i = f_\theta(\mathbf{x}_i)$, and $\boldsymbol{\mu}_j$ is the center of cluster $j$. The target distribution $\mathbf{P}$ is defined as:

$$p_{i,j} = \frac{q_{i,j}^2/f_j}{\sum_{j'}(q_{i,j'}^2/f_{j'})}, \tag{4}$$

where $f_j = \sum_i q_{i,j}$ are the soft cluster frequencies. Here, $q_{i,j}^2$ is used to strengthen predictions on data points that are assigned with high confidence. Simultaneously, a division by $f_j$ should prevent large clusters from dominating the loss.

Putting everything together, we receive:

$$L_C(\theta, \mathbf{M}) = KL(P||Q) = \sum_i \sum_j p_{i,j} \log(\frac{p_{i,j}}{q_{i,j}}). \tag{5}$$

In the case of DEC, $\lambda_1 = 0$ and $\lambda_2 = 1$ during the clustering optimization. For IDEC, $\lambda_1 = 1$ and $\lambda_2 = 0.1$ to prevent a distorted embedding.

**DCN:** The optimization strategy of DCN (Yang et al., 2017) is based on the classical $k$-Means (Lloyd, 1982) algorithm. First, the network parameters $\theta$ are updated using $L(\theta, \mathbf{M})$, where

$$L_C(\theta, \mathbf{M}) = \frac{1}{2} \sum_i ||\mathbf{z}_i - \boldsymbol{\mu}_{h(\mathbf{z}_i)}||_2^2 \tag{6}$$

and $h(\mathbf{z}_i)$ returns the cluster $\mathbf{z}_i$ is assigned to. Afterward, each sample $\mathbf{z}_i$ is assigned to its closest cluster center. Last, the cluster centers are updated by calculating:

$$\boldsymbol{\mu}_{h(\mathbf{z}_i)}^{\text{updated}} = \boldsymbol{\mu}_{h(\mathbf{z}_i)} - \frac{1}{c_{h(\mathbf{z}_i)}}(\boldsymbol{\mu}_{h(\mathbf{z}_i)} - \mathbf{z}_i). \tag{7}$$

Here, $c_j$ counts the total number of samples already assigned to cluster $j$. This update strategy is inspired by the SGD-$k$-Means algorithm (Bottou & Bengio, 1994). Finally, the process repeats by optimizing the embedding using Eq. 6 with the newly obtained cluster parameters.

B.2 SIMCLR

SimCLR (Chen et al., 2020) is a contrastive learning method designed to discriminate among pairs of augmented samples. In its pretraining phase, the encoder $f_\theta$ processes a batch of $N$ samples, each augmented twice, resulting in a batch size of $2N$. This yields embeddings $\mathbf{h}_i = f_\theta(\mathbf{x}_i)$. Instead of explicitly mining negatives, SimCLR treats all augmented samples not belonging to the anchor as negative samples. Before the contrastive loss is applied, the encoded samples are mapped into another latent space $\mathbf{z}_i = g(\mathbf{h}_i)$ by a projector network. SimCLR then applies the InfoNCE loss (van den Oord et al., 2018):

$$L_{i,j} = -\log \frac{\exp(\text{sim}(\mathbf{z}_i, \mathbf{z}_j)/\tau)}{\sum_{k=1}^{2N} \mathbb{1}_{[k \neq i]} \exp(\text{sim}(\mathbf{z}_i, \mathbf{z}_k)/\tau)}, \tag{8}$$

where $\text{sim}(\mathbf{z}_i, \mathbf{z}_j)$ denotes the cosine similarity between two projected embeddings, and $\mathbb{1}_{[k \neq i]}$ is 1 if $k \neq i$. Essentially, contrastive learning fosters a latent space wherein the features of augmented versions of the same image are close while being distanced from others, thereby promoting a cluster-friendly latent space (Wang & Isola, 2020).

Table 3: Information regarding the benchmark datasets.

| Dataset | Type | # Samples | # Features | # Classes |
|---|---|---|---|---|
| Optdigits | Grayscale | 5,620 | $8 \times 8$ | 10 |
| GTSRB | Color | 7,860 | $32 \times 32 \times 3$ | 5 |
| USPS | Grayscale | 9,298 | $16 \times 16$ | 10 |
| CIFAR10 | Color | 60,000 | $32 \times 32 \times 3$ | 10 |
| CIFAR100-20 | Color | 60,000 | $32 \times 32 \times 3$ | 20 |
| MNIST | Grayscale | 70,000 | $28 \times 28$ | 10 |
| FMNIST | Grayscale | 70,000 | $28 \times 28$ | 10 |
| KMNIST | Grayscale | 70,000 | $28 \times 28$ | 10 |

### B.3 RESET METHODS IN OTHER FIELDS

Reset methods have been used in other fields to great success. For example, it has been shown that soft resets —named *Shrink and Perturb*— may be able to alleviate poor generalization performance in the pretrain + fine-tune setting in supervised learning (Ash & Adams, 2020). This phenomenon has been dubbed the *generalization gap* by Berariu et al. (2021).

A striking application of resetting has developed in the field of deep Reinforcement Learning (RL): Here, resets have been used to improve the data efficiency of algorithms by mitigating *plasticity loss* (Nikishin et al., 2022; D'Oro et al., 2023; Schwarzer et al., 2023). Loss of plasticity refers to the loss of a network's learning ability over the course of training (Berariu et al., 2021; Dohare et al., 2021). It is particularly prevalent in RL due to the non-stationarity induced by bootstrapped targets in deep value-based algorithms.

As our disentanglement ablation study in Figure 10 shows, deep clustering algorithms do not suffer from a loss of plasticity. Instead, the resets help avoid premature convergence to sub-optimal local minima by facilitating the exploration of multiple clustering solutions. When BRB commits to a clustering solution, the resets ensure that it is robust with respect to the structured noise induced by the resets.

The different reset strategy our method employs highlights this: Our method never applies Shrink and Perturb to the embedding layer, as this could lead to a collapse of deep clustering performance. On the other hand, deep RL algorithms often fully reset the last layers completely, as plasticity loss is commonly believed to be concentrated there (Berariu et al., 2021; D'Oro et al., 2023).

## C DATASETS

We evaluate our approach by analyzing the results regarding eight image datasets commonly used in machine learning. The information concerning these datasets is summarized in Tab. 3.

The grayscale image datasets Optdigits (Alpaydin & Kaynak, 1998), USPS (Hull, 1994), and MNIST (LeCun et al., 1998) contain images of handwritten digits (0-9), with each dataset having a different resolution. FMNIST (Xiao et al., 2017) contains grayscale images of ten articles from the Zalando online store, and KMNIST (Clanuwat et al., 2018) is composed of grayscale images showing ten Japanese Kanji characters.

The color dataset GTSRB (Stallkamp et al., 2012) contains images of 43 German traffic signs. We use a subset of the data containing the signs 'Speed limit 50', 'Priority road', 'No entry', 'General caution', and 'Ahead only' (as proposed in (Leiber et al., 2021) - 'Priority road', 'No entry', and 'General caution' were given as 'Right of way', 'No passing', and 'Attention' here) and unify the sizes of the images. CIFAR10 (Krizhevsky et al., 2009) is composed of color images showing ten different objects with various backgrounds. The CIFAR100-20 dataset uses a course-grained grouping of the 100 classes of the CIFAR100 dataset to summarize the fine-grained classes within related super-classes.

All datasets are preprocessed by performing a channel-wise-z-transformation, resulting in a mean of zero and a unit variance for each color channel. In addition, histogram normalization is conducted for GTSRB due to the low contrast.

# D   METRICS

In the following, we define the clustering evaluation metrics that are used throughout our work. More detailed descriptions and examples can be found in Murphy (2022).

## D.1   CLUSTERING ACCURACY

Clustering accuracy (ACC) (Yang et al., 2010) measures the fraction of correctly assigned cluster labels over the dataset compared to the ground truth labels:

$$\text{ACC} = \max_{\text{map}} \frac{\sum_{i=1}^{n} \delta(y_i, \text{map}(\hat{y}_i))}{n}. \tag{9}$$

Here $y_i$ is the ground-truth label of data point $i$, $\hat{y}_i$ the predicted label of sample $i$, and $\delta$ denotes the Kronecker-Delta. As the labels for the found clustering might deviate from the ground truth labels, we require the mapping function map that tests each possible mapping of labels. We use the Hungarian method (Kuhn, 1955) to identify the best mapping.

## D.2   ARI

Adjusted Rand Index (ARI) (Hubert & Arabie, 1985) is a statistical measure that generalizes the notion of accuracy for clustering and additionally adjusts for chance. It provides a score $\text{ARI} \in [-1, 1]$. An ARI score of 1 signifies agreement, $-1$ indicates disagreement, and a score of 0 suggests agreement by chance. ARI is defined as:

$$\text{ARI} := \frac{\text{R} - \mathbb{E}_{\hat{Y} \sim \text{HGeom}(\mathbf{k}, n)}[R]}{1 - \mathbb{E}_{\hat{Y} \sim \text{HGeom}(\mathbf{k}, n)}[R]}, \tag{10}$$

where $\text{R}$ denotes the Rand index, which corresponds to ACC in case the ground truth class labels are used as reference clustering. ARI adjusts for randomness by using the expected value of the Rand index $\mathbb{E}_{\hat{Y} \sim \text{HGeom}(\mathbf{k}, n)}[R]$ under a generalized Hypergeometric distribution $\text{HGeom}(\mathbf{k}, n)$ with $\mathbf{k} = (k_1, \ldots, k_n)$ to distribute the $n$ samples across the $k$ classes with uniform random probability.

## D.3   NMI

The Normalized Mutual Information (NMI) (Kvalseth, 1987) calculates the Mutual Information between two sets of labels and scales the results between 0 and 1.

$$\text{NMI}(C, \hat{C}) := \frac{2I(C, \hat{C})}{H(C) + H(\hat{C})}, \tag{11}$$

where $C = \{C_1, \ldots, C_k\}$ is the ground truth partition and $\hat{C} = \{\hat{C}_1, \ldots, \hat{C}_{k'}\}$ are the predicted clusters. $I$ denotes mutual information between two sets, whereas $H(X)$ corresponds to the entropy of a probability distribution $X$.

## D.4   INTER- & INTRA- CLASS DISTANCE

Lehner et al. (2024) uses the silhouette score Rousseeuw (1987) of the l2 normalized embeddings to measure the separation of ground truth clusters in the embedded space of neural networks. The silhouette score is the relation between the intra- and inter-class distances that we use in this work.

The intra-class distance (intra-CD) of a sample $\mathbf{x}_i \in C_l$ is defined as the mean distance of the sample to other samples $\mathbf{x}_j$ in the *same cluster* $C_l$:

$$d_{\text{intra}}(\mathbf{x}_i) := \frac{1}{|C_l| - 1} \sum_{\mathbf{x}_j \in C_l} d(\mathbf{x}_i, \mathbf{x}_j). \tag{12}$$

The inter-class distance (inter-CD) of a sample $\mathbf{x}_i \in C_l$ is the mean distance of the sample to instances in the *next closest cluster* $C_{m \neq l}$:

$$d_{\text{inter}}(\mathbf{x}_i) := \min_{m \neq l} \left( \frac{1}{|C_m|} \sum_{\mathbf{x}_j \in C_m} d(\mathbf{x}_i, \mathbf{x}_j) \right). \tag{13}$$

## E  IMPLEMENTATION DETAILS AND HYPERPARAMETERS

Our codebase builds on top of the open-source ClustPy-codebase (Leiber et al., 2023), which provides Pytorch implementations of the centroid-based deep clustering (DC) algorithms used in our work. We use the Adam optimizer (Kingma & Ba, 2015) due to its widespread use and strong performance on a wide range of deep learning problems. For DEC and IDEC, the centroids are optimizable parameters. Therefore, we also reset their momentum after each weight reset to prevent divergent gradients. Due to the high learning rates used in our contrastive learning experiments (cf. Table 7), we rarely encounter single gradient steps with very high gradient magnitudes. Therefore, we clip the $\ell_2$ gradient norms at a value of $500$. For the AE, the latent dimension is set to the dataset's number of classes, a common practice in DC. Additionally, we reset the projector network for contrastive learning with each weight reset of BRB.

We have to slightly adapt the objective functions of DEC, IDEC, and DCN to handle augmentations in a meaningful way. An essential constraint is that the cluster assignments of a sample should not change due to the applied augmentations. Therefore, the augmented sample $\mathbf{x}_i^A = \text{aug}(\mathbf{x}_i)$ must always receive the same assignments $s_{i,j}$ as the original sample $\mathbf{x}_i$. We denote $\mathbf{z}_i^A = f_\theta(\mathbf{x}_i^A)$ as the encoded version of $\mathbf{x}_i^A$. In general, we use the average of the loss with respect to the original sample $\mathbf{x}_i$ and the loss with respect to the augmented sample $\mathbf{x}_i^A$.

First, we adjust $L_{SSL}$ (here shown for the AE):

$$L_{SSL}^A = \frac{1}{2} \left( L_{SSL}(\theta, \mathbf{x}_i) + L_{SSL}(\theta, \mathbf{x}_i^A) \right). \tag{14}$$

Next, we update the clustering loss $L_C$ regarding **DEC** and **IDEC**:

$$L_C^A = \frac{1}{2} \sum_i \sum_j \left( p_{i,j} \log(\frac{p_{i,j}}{q_{i,j}}) + p_{i,j} \log(\frac{p_{i,j}}{q_{i,j}^A}) \right), \tag{15}$$

where

$$q_{i,j}^A = \frac{(1 + ||\mathbf{z}_i^A - \boldsymbol{\mu}_j||_2^2)^{-1}}{\sum_{j'} (1 + ||\mathbf{z}_i^A - \boldsymbol{\mu}_{j'}||_2^2)^{-1}}. \tag{16}$$

Note that $p_{i,j}$ remains the same compared to the DEC/IDEC version without augmentation.

Last, we adjust $L_C$ for **DCN**:

$$L_C^A = \frac{1}{4} \sum_i \left( ||\mathbf{z}_i - \boldsymbol{\mu}_{h(\mathbf{z}_i)}||_2^2 + ||\mathbf{z}_i^A - \boldsymbol{\mu}_{h(\mathbf{z}_i)}||_2^2 \right) \tag{17}$$

For contrastive learning, which learns a representation that is invariant to the augmentations, we only use augmented samples during pretraining and deep clustering.

### E.1 SELF-LABELING IMPLEMENTATION DETAILS

Many deep clustering methods have two stages: pretraining and joint deep clustering. Self-labeling (Gansbeke et al., 2020) can be used after joint deep clustering to further improve clustering performance as an additional third stage. Self-labeling uses ideas from self-training in semi-supervised learning to only train on samples that have already been assigned with high certainty to a cluster. Self-labeling does not work out of the box with centroid-based deep clustering methods as it assumes a classification head to discriminate between the pseudo-labels learned from the clustering. To solve this problem, we follow the training strategy of (Qian, 2023), which successfully applied self-labeling to their centroid-based deep clustering method SeCu by introducing a short warmup phase. This warmup phase trains a classification head based on the clustering labels obtained from the centroid-based clustering objective and is then followed by the self-labeling phase[3].

### E.2 HYPERPARAMETER SENSITIVITY ANALYSIS

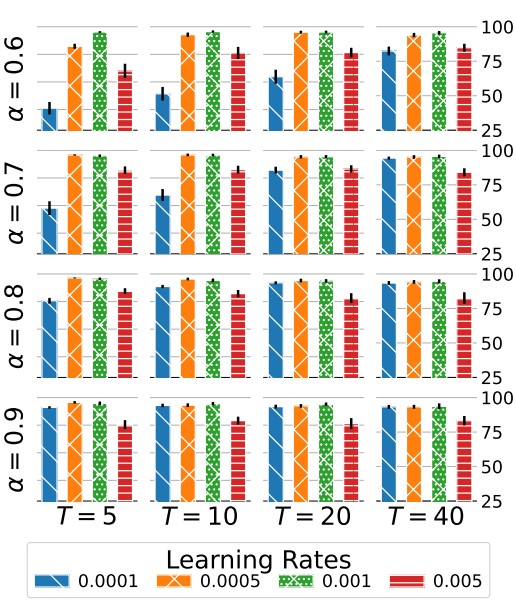

Figure 8: **Hyperparameter sensitivity analysis.** Clustering accuracy (averaged over DEC/IDEC/DCN and five seeds per setting on MNIST) for BRB's reset factor $\alpha$, reset interval $T$ and different deep clustering learning rates. Large ($\alpha \in \{0.6, 0.7\}$) and frequent ($T \in \{5, 10\}$) resets can destroy cluster structure (lower accuracy), whereas moderate values ($T \in \{20, 40\}$, $\alpha \in \{0.8, 0.9\}$) perform consistently well.

In Figure 8, we provide a detailed sensitivity analysis of BRB over the learning rate (LR), the reset interval $T$, and the reset factor $\alpha$. We report averages over 15 runs (five seeds per DEC, IDEC, and DCN) for each setting on MNIST, resulting in 960 runs in total. For the AE pretraining, we use an LR of 0.001 and, therefore, select LRs that are higher (0.005), equal (0.001), and lower (0.0005, 0.0001) than the pretraining LR for DC. Note that usually, only LRs that are lower than the pretraining LR are considered in DC. In Figure 8, the lower and higher end of LRs (0.0001, 0.005) suffer more from frequent ($T \in \{5, 10\}$) and strong ($\alpha \in \{0.6, 0.7\}$) resets. Intuitively this makes sense: The more often we reset with a high reset factor, the more likely it is to destroy useful cluster information. The lower and higher LRs exacerbate this by either recovering too slowly or too quickly, whereas the moderate LRs (0.0005, 0.001) are still robust. An exception occurs for $\alpha = 0.6$ with $T = 5$, where the setting with LR 0.0005 loses some performance. In general, we see that the moderate LRs (0.0005 and 0.001) remain stable over all other settings. As a rule of thumb, choosing a moderate reset value between 0.8 and 0.9 with less frequent resets ($T \in \{20, 40\}$) and an LR that is equal to or slightly lower than the pretraining LR performs well in many settings. This is why for all main experiments, we choose an LR of 0.001 and set $\alpha = 0.8$ with $T = 20$.

---

[3]We used the public repository of SCAN at `https://github.com/wvangansbeke/Unsupervised-Classification`

## E.3 ADDITIONAL HYPERPARAMETER SENSITIVITY EXPERIMENTS

Figures 9a and 9b show hyperparameter sensitivity matrix plots for the datasets USPS and OPT-DIGITS. Compared to MNIST, both USPS and OPTDIGITS suffer more from frequent or strong resets. For example, setting $\alpha = 0.6$ and $T = 5$ with a learning rate of $0.001$ yields performance well below $50\%$ clustering accuracy on USPS and completely fails to learn on OPTDIGITS. In contrast, the same settings on MNIST (cf. Figure 8) result in similar clustering accuracy values as the ones obtained by the settings we use in the main paper, namely $\alpha = 0.8$, $T = 20$, and learning rate $0.001$. Interestingly, both USPS and OPTDIGITS seem to benefit from using a higher learning rate of $0.05$ (red bars), whereas increasing the learning rate to this level is harmful on MNIST. We conclude that (i) the best-performing ranges of hyperparameters differ depending on the dataset, and (ii) there may be trade-offs between hyperparameters for different datasets. The hyperparameters we used for all our main experiments ($\alpha = 0.8$, $T = 20$, learning rate $0.001$) are within a range of settings that perform well on a variety of datasets.

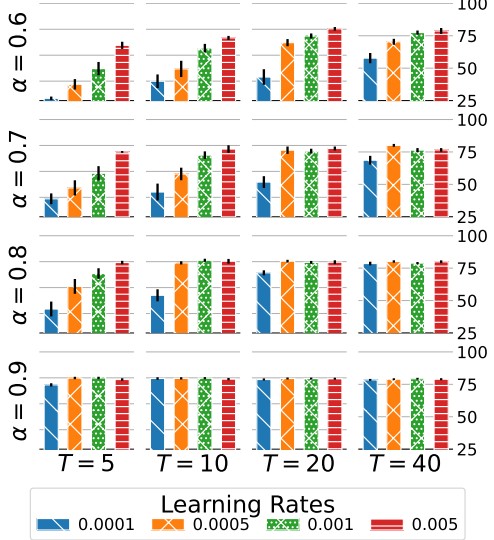

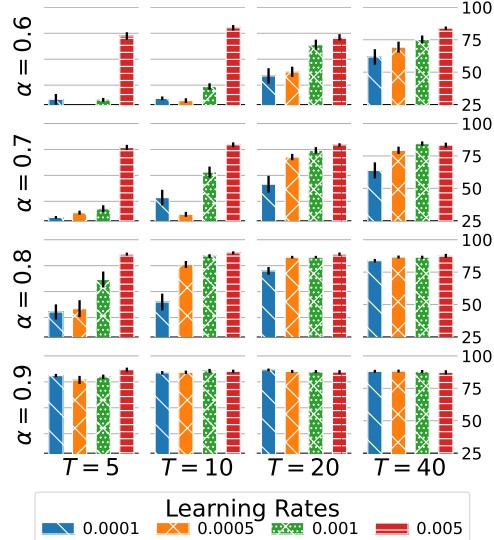

(a) **Hyperparameter sensitivity analysis on USPS.** Clustering accuracy (averaged over DEC/IDEC/DCN and five seeds per setting) for BRB's reset factor $\alpha$, reset interval $T$, and different deep clustering learning rates.

(b) **Hyperparameter sensitivity analysis on OPT-DIGITS.** Clustering accuracy (averaged over DEC/IDEC/DCN and five seeds per setting) for BRB's reset factor $\alpha$, reset interval $T$, and different deep clustering learning rates.

Figure 9: Hyperparameter sensitivity analysis for USPS and OPTDIGITS.

### E.4 COMPUTATIONAL RESOURCES

We primarily utilized internal servers to generate our results with grayscale datasets. For this paper, we had access to

- $2\times$ Intel Xeon Silver 4214R CPU @ 2.40GHz server with 48 cores and 2 Nvidia A100 GPUs with 40GB of VRAM each
- $2\times$ Intel Xeon Gold 6326 CPU @ 2.90GHz server with 64 cores and 1 NVIDIA A100 GPU with 80GB of VRAM

These machines were also used to run preliminary experiments and to identify broad working ranges for the most important hyperparameters.

For the experiments with color datasets and contrastive learning, we had access to 1000h compute hours per month on a SLURM supercomputing cluster. A node in this supercomputer uses an AMD EPYC 7452 processor with 128 cores and 1 NVIDIA A100 GPU with 80GB of VRAM. We also used these resources to tune hyperparameters for SimCLR (Chen et al., 2020) pretraining.

The average wall clock time for one run is summarized in Table 4. We identify two main factors contributing to the runtime of our method. First, the number of samples in the dataset (see Table 3), where, e.g., MNIST variants take substantially longer to run than USPS or OPTDIGITS. Second, the number of CPU cores we could utilize to parallelize data augmentations.

Table 4: Average runtime of our experiments.

| Dataset | Approximate Average runtime |
|---|---|
| MNIST, FMNIST, KMNIST | 4h |
| GTSRB | 20m |
| USPS | 15m |
| OPTDIGITS | 8m |
| CIFAR10 | 5h |
| CIFAR100-20 | 5h |

### E.5 HYPERPARAMETERS FOR GRAYSCALE DATASETS

Table 5 lists the hyperparameters used for our experiments on grayscale datasets. These consist of OPTDIGITS (Alpaydin & Kaynak, 1998), USPS (Hull, 1994), MNIST (LeCun et al., 1998), FMNIST (Xiao et al., 2017), and KMNIST (Clanuwat et al., 2018). We tune the parameters for $\alpha \in \{0.6, 0.7, 0.8, 0.9\}$ and $T \in \{10, 20, 40\}$ once on MNIST and re-use them for all grayscale datasets.

| Parameter | Value |
|---|:---:|
| GENERAL TRAINING | |
|     Batch size | 256 |
|     Learning rate | $1e-3$ |
| PRETRAINING | |
|     Epochs | 250 |
|     Augmentation | False |
| CLUSTERING | |
|     Epochs | 400 |
|     Augmentation | True |
|     DCN cluster loss | 0.025 |
|     IDEC cluster loss | 0.1 |
| AUGMENTATION | |
|     `Resize` | $(28, 28)$ |
|     `RandomAffine` | $\deg = \pm 16, \text{transl} = 0.1,$ $\text{shear} = \pm 8$ |
| ARCHITECTURES | |
|     Feed Forward | $D$-1024-512-256-$d$ |
|     BatchNorm | False |
|     Embedding size | # Clusters |
| OPTIMIZER | Adam |
|     Momentum $\beta_1$ | 0.9 |
|     Momentum $\beta_2$ | 0.999 |
|     Weight decay | 0 |
| BRB | |
|     Subsample size | 10000 |
|     $\alpha$ (Reset interpolation factor) | 0.8 |
|     $T$ (Reset interval) | 20 |
|     Reset embedding | False |
|     Reset center momentum | True |

Table 5: BRB hyperparameters for grayscale datasets.

### E.6 Hyperparameters for Color Datasets with Reconstruction Pretraining

Our hyperparameters used for the dataset GTSRB (Stallkamp et al., 2012) are in Table 6. We use the same hyperparameters as for the grayscale datasets except for the data augmentations.

| Parameter | Value |
|---|---|
| GENERAL TRAINING | |
| Batch size | 256 |
| Learning rate | $1e-3$ |
| PRETRAINING | |
| Epochs | 250 |
| Augmentation | False |
| CLUSTERING | |
| Epochs | 400 |
| Augmentation | True |
| DCN cluster loss | 0.025 |
| IDEC cluster loss | 0.1 |
| AUGMENTATIONS | |
| RandomResizedCrop | $(32, 32)$ |
| ColorJitter | $\text{bright} = 0.4, \text{contrast} = 0.4,$ $\text{sat} = 0.2, \text{hue} = 0.1, p = 0.8$ |
| RandomGrayscale | $p = 0.2$ |
| RandomHorizontalFlip | $p = 0.5$ |
| RandomSolarize | $\text{thresh} = 0.5, p = 0.2$ |
| ARCHITECTURE | |
| Feed Forward | $D$-1024-512-256-$d$ |
| Embedding size | # Clusters |
| BatchNorm | False |
| OPTIMIZER | Adam |
| Momentum $\beta_1$ | 0.9 |
| Momentum $\beta_2$ | 0.999 |
| Weight decay | False |
| BRB | |
| Subsample size | 10000 |
| $\alpha$ (Reset interpolation factor) | 0.8 |
| $T$ (Reset interval) | 20 |
| Reset embedding | False |
| Reset center momentum | True |

Table 6: BRB hyperparameters for GTSRB (Stallkamp et al., 2012).

### E.7 HYPERPARAMETERS FOR COLOR DATASETS WITH CONTRASTIVE LEARNING

We conduct experiments with contrastive learning as auxiliary tasks on CIFAR10 (Krizhevsky et al., 2009) and CIFAR100-20. Some parameters depend on the dataset and algorithm in this setup, we therefore report them separately in Table 7. For self-labeling (Gansbeke et al., 2020) we use the same configs as specified in their public repository[4], except for the differing hyperparameters specified at the bottom of Table 7.

Table 7: BRB hyperparameters when using contrastive learning as auxiliary task on CIFAR10 (Krizhevsky et al., 2009) and CIFAR100-20.

| Parameter | CIFAR10 | CIFAR100-20 |
|---|---|---|
| GENERAL TRAINING | | |
| Batch size | 512 | 512 |
| PRETRAINING | | |
| Epochs | 1000 | 1000 |
| Learning rate | $3e-3$ | $3e-3$ |
| Weight decay | 1e-4 | 1e-4 |
| Augmentation | True | True |
| CLUSTERING | | |
| Learning rate | 1e-3 (DEC), 3e-3 (IDEC), 1e-2 (DCN) | 1e-3 (DEC), 3e-3 (IDEC), 1e-2 (DCN) |
| Epochs | 1000 | 1000 |
| Augmentation | True | True |
| **DCN** cluster loss weight | $1e-4$ | $1e-4$ |
| **IDEC** cluster loss weight | 1.0 | 1.0 |
| AUGMENTATIONS | | |
| RandomResizedCrop | $(32, 32)$ | $(32, 32)$ |
| ColorJitter | bright $= 0.4$, contrast $= 0.4$, sat $= 0.2$, hue $= 0.1, p = 0.8$ | bright $= 0.4$, contrast $= 0.4$, sat $= 0.2$, hue $= 0.1, p = 0.8$ |
| RandomGrayscale | $p = 0.2$ | $p = 0.2$ |
| RandomHorizontalFlip | $p = 0.5$ | $p = 0.5$ |
| RandomSolarize | thresh $= 0.5, p = 0.2$ | thresh $= 0.5, p = 0.2$ |
| ARCHITECTURE | | |
| CNN | ResNet-18 | ResNet-18 |
| Feed Forward | $D$-512-256-$d$ | $D$-512-256-$d$ |
| $d$ (Embedding size) | 128 | 128 |
| Projector depth | 1 | 1 |
| Projector width | 2048 | 2048 |
| Softmax Temperature | 0.5 | 0.5 |
| BatchNorm | True | True |
| OPTIMIZER | Adam | Adam |
| Momentum $\beta_1$ | 0.9 | 0.9 |
| Momentum $\beta_2$ | 0.999 | 0.999 |
| **DEC/IDEC** Weight decay | False | False |
| **DCN** Weight decay | False | False |
| BRB | | |
| Subsample size | 10000 | 10000 |
| $\alpha$ (Reset interpolation factor) | | |
| MLP reset factor | 0.7 | 0.7 |
| ResNet Block 4 reset factor | 0.9 | 0.9 |
| $T$ (Reset interval) | 10 | 10 |
| Reset embedding | False | False |
| Reset center momentum | True | True |
| SELF-LABELING | | |
| We use the same parameters as (Gansbeke et al., 2020) except for the following: | | |
| Learning rate | 0.001 | 0.001 (DEC & IDEC) 0.00025 (DCN) |
| Warmup Epochs | 20 | 20 |
| OPTIMIZER | Adam | Adam |
| Momentum $\beta_1$ | 0.9 | 0.9 |
| Momentum $\beta_2$ | 0.999 | 0.999 |
| Weight decay | False | False |

---

[4]The config for CIFAR100-20 can be found at `https://github.com/wvangansbeke/Unsupervised-Classification/blob/master/configs/selflabel/selflabel_cifar20.yml` and the one for CIFAR10 is in the same folder denoted as *selflabel_cifar10.yml*.

# F  ABLATION FOR NEW CLUSTERING TARGETS

In this section, we describe the setups of our disentanglement and noise experiments from the main paper. We then provide a more detailed analysis of the results.

## F.1  EXPERIMENT SETUP FOR THE DISENTANGLEMENT ABLATION

With the disentanglement ablation, our objective is to isolate the impact of altered cluster labels resulting from resets from the resets themselves. For this, we modify the pseudocode of Algorithm 1 as follows: Initially, we reset the weights of a duplicate of the DC algorithm's model in the first and second steps, utilizing it to produce new embeddings. These embeddings are then used to derive new cluster labels while keeping the existing centroids unchanged. Subsequently, new centroids are computed using embeddings from the unaltered model but with the newly generated labels. The modifications are outlined in Algorithm 2.

---

**Algorithm 2** PyTorch-style pseudo-code of Disentangled BRB

```
# model: neural network
# dc: deep clustering method
# alpha: BRB reset factor (0 <= alpha <= 1)
# recluster_algorithm: algorithm for reclustering
# T: BRB reset interval
# optimizer: optimizer to be used, default : Adam
# subsample_size: size of sample used for reclustering

# dc_method training loop
for epoch in epochs:
    if epoch % T == 0 and epoch > 0:
        # perform reset on copy
        reset_copy = reset_model_weights(copy(model), alpha)
        # embed data with reset copy of model
        # optional: subsampling can be used
        noise_emb = embed_data(reset_copy, loader, subsample_size)
        emb = embed_data(model, loader, subsample_size)
        # new labels
        labels = predict_labels(dc.method.centroids, model, noise_emb)
        # use embedding of unperturbed model
        dc.centroids = recluster_algorithm(emb, labels)
        if dc.centroids.has_momentum():
            # reset momentum of learnable centroids
            reset_momentum(optimizer, dc.centroids)

    # load a minibatch x
    for x in loader:
        # perform deep clustering update steps
        dc.update(x)
```

---

## F.2  EXPERIMENT SETUP FOR THE NOISE ABLATION

BRB employs soft resets to introduce structured noise into the optimization process of the DC algorithm. While this approach offers the advantage of impacting both the deep learning components and clustering, one might hypothesize that simply adding noise to generate a new clustering could suffice to escape sub-optimal local minima. We explore this hypothesis through an ablation study where we add noise to the embeddings instead of resetting the network's weights. To ensure bespoke distortions in the latent space, we first normalize the noise vector to match the norm of the embeddings. We then add a fraction of that vector, as described in Equation 18:

$$\forall i \in \{1, \ldots, n\} : \tilde{\mathbf{h}}_i = \mathbf{h}_i + \beta \boldsymbol{\epsilon}, \quad \text{with } \|\boldsymbol{\epsilon}\| = \|\mathbf{h}_i\| \tag{18}$$

Here, $\boldsymbol{\epsilon}$ denotes independent and identically distributed noise across dimensions and samples. $\beta \in \mathbb{R}^+$ represents a positive scaling factor. Empirically, we find that Gaussian noise with scaling factor $\beta = 0.3$ achieves a balance between inducing label changes and minimizing performance degradation in later training stages. Following noise addition, we recluster using the perturbed embeddings while continuing training on the unmodified network.

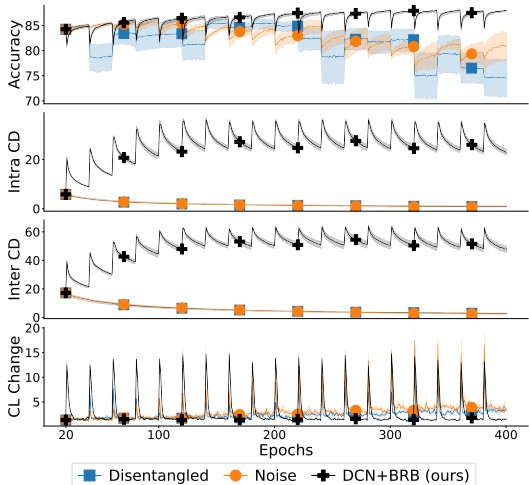

Figure 10: **Importance of resets.** Clustering accuracy on OPTDIGITS drops considerably if only targets are changed without resetting the embedding (*Disentangled*). Similarly, adding Gaussian noise to the embedding is insufficient to change cluster structure (*Noise*).

### F.3    DETAILED ANALYSIS OF THE DISENTANGLEMENT AND NOISE ABLATION

To disentangle the effect of weight resetting from the influence of new clustering targets obtained through reclustering on the post-reset embedding, we perform two ablation studies: First, the *Disentangled* variant compares BRB with a modified algorithm utilizing labels generated from a perturbed embedding, achieved by resetting a duplicate encoder. This is followed by reclustering with $k$-Means on the perturbed encoder's embedded space without changing the original network. For the second experiment, we add noise to the embedded space before reclustering. The exact experiment setups are described in the previous sections.

The results in Figure 10 show that for both modifications of BRB, the performance degrades, whereas BRB maintains the network's ability to fit new targets due to the reset and yields strictly better results. The intra-CD and inter-CD of both the *Disentangled* and *Noise* versions remain virtually unchanged compared to the baseline, indicating that a persistent change in the embedded space is required to enable enhanced adaptation. In terms of CL change, the modified algorithms either fail to induce the required changes to explore new clustering solutions (*Disentangled*) or generate too large label changes late in training (*Noise*). Both negatively affect clustering performance. In summary, this lets us conclude that *structured, persistent perturbations* induced by our soft resets are a necessary condition for subsequent improvements through the reclustering step of BRB.

## G    RUNTIME ANALYSIS

### G.1    RUNTIME COMPLEXITY

BRB consists of three components: parameter perturbations, reclustering, and momentum resets. We will first give the runtime complexity of these components and then verify these findings empirically. In summary: Our analysis reveals that the reclustering step of BRB adds the most overhead, but compared to the total training time, the impact of BRB is modest, for instance, resulting in a 5.38% increase in total training time for our experiments for IDEC on MNIST.

**Parameter perturbations.**    The complexity is $O(L \cdot D^2)$, where $L$ is the number of layers that are to be reset and $D$ are the most neurons across all layers that are to be reset. Note that $L$ is usually small as we do not reset the full network for larger models and that the bound does not depend on the size of the dataset.

**Reclustering.** Here, we must analyze two stages. The first stage consists of forward passes to embed the subsampled data points. We assume that our model is an $M$-layer feedforward network to quantify runtime complexity. The second stage is the reclustering algorithm. Combined, we obtain a complexity of $O(N \cdot M \cdot D^3 + I \cdot k \cdot N)$, where $N$ is the number of subsampled data points for reclustering, $M$ is the total number of layers of the network, $k$ the number of clusters and $I$ the maximum number of clustering iterations (we use the sklearn default of $I = 300$). We stress that while the first term seems prohibitively expensive, it is usually heavily parallelized through batched GPU implementations of deep learning frameworks such as PyTorch.

**Momentum resets.** We just set the centroid momentum tensors to zero, which requires negligible compute.

G.2 EMPIRICAL RESULTS

All results discussed in this section are generated using an Intel Xeon Silver 4214R CPU @ 2.40GHz server with 12 physical and 48 virtual cores and 2 Nvidia A100 GPUs with 40GB of VRAM each.

For our empirical investigations, we measure the time for each BRB component individually in seconds (10 seed averages) and compare them with the time of one epoch of deep clustering training. Table 8 shows time estimates for IDEC+BRB on the MNIST dataset, including the constant (as a function of the number of samples) and negligible overhead for the perturbations. We separately report the time for embedding and clustering, which together comprise the reclustering phase.

Table 8: Runtime in seconds for IDEC+BRB on MNIST with different subsample sizes.

| Subsample size | Reset time | Embedding time | Cluster time | Momentum reset time | BRB time | Total runtime | BRB % of total runtime |
|---|---|---|---|---|---|---|---|
| 700 (1%) | 0.007 | 0.639 | 0.063 | 0.001 | 0.710 | 5.025 | 14.131 |
| 7000 (10%) | 0.006 | 0.630 | 0.337 | 0.001 | 0.975 | 5.417 | 17.991 |
| 17500 (25%) | 0.007 | 0.710 | 0.866 | 0.001 | 1.583 | 5.872 | 26.964 |
| 35000 (50%) | 0.007 | 0.803 | 1.862 | 0.001 | 2.672 | 7.063 | 37.830 |
| 70000 (100%) | 0.007 | 1.068 | 3.647 | 0.001 | 4.724 | 9.113 | 51.835 |

For our experiments, we apply BRB every 20th epoch, and thus, the runtime cost of BRB is distributed over 20 epochs. In this case, reclustering on the full dataset leads to an increase in the total runtime of the clustering phase of 5.38%, which is reduced to 1.1% when only 10% of the samples are used. We calculate these numbers using the formula $\frac{\text{time(BRB)}}{\text{time(No BRB)}} = \frac{20*(\text{Total runtime}-\text{BRB time})+\text{BRB time}}{20*(\text{Total runtime}-\text{BRB time})}$. In our opinion, the overhead added by BRB is minor compared to its benefits and could, for instance, be further reduced by caching embeddings. Note that while the time increases with more frequent application of BRB, subsampling can further reduce the runtime at a minimal cost in clustering performance, as shown in Table 9 below:

Table 9: MNIST ARI for different clustering algorithms and subsample sizes.

| Subsample size | IDEC | DEC | DCN |
|---|---|---|---|
| 700 (1%) | $91.84 \pm 1.99$ | $90.94 \pm 2.37$ | $89.19 \pm 1.59$ |
| 7000 (10%) | $93.04 \pm 1.90$ | $90.04 \pm 2.23$ | $91.99 \pm 0.23$ |
| 17500 (25%) | $93.06 \pm 1.90$ | $89.99 \pm 2.43$ | $91.97 \pm 0.22$ |
| 35000 (50%) | $93.05 \pm 1.90$ | $90.49 \pm 1.95$ | $92.08 \pm 0.24$ |
| 70000 (100%) | $93.02 \pm 1.90$ | $91.60 \pm 1.93$ | $92.00 \pm 0.21$ |

In Section 5, we state that "any centroid-based method" can be used for the reclustering stage of BRB. In Table 10, we analyze four different reclustering algorithms in terms of runtime: $k$-Means (Lloyd, 1982), $k$-Means++ initialization (Arthur & Vassilvitskii, 2007), $k$-Medoids (Rdusseeun & Kaufman, 1987) and the expectation-maximization (EM) algorithm (Dempster et al., 1977). To calculate the percentage-based timings, we use the same formula as above. Unsurprisingly, EM takes the longest

to run as it has to calculate full covariance matrices for all components of the fitted mixture model. $k$-Means++-init is the fastest, hinting at a possible strategy to speed up BRB. Our choice of $k$-Means provides simplicity while having intermediate speed.

Table 10: Runtime in seconds for IDEC+BRB on MNIST with different reclustering algorithms.

| Clustering method | Reset time | Embedding time | Cluster time | Momentum reset time | BRB time | Total runtime | BRB % of total runtime |
|---|---|---|---|---|---|---|---|
| EM | 0.006 | 0.465 | 8.944 | 0.001 | 9.416 | 10.239 | 91.968 |
| $k$-Means | 0.006 | 0.429 | 0.500 | 0.001 | 0.936 | 01.725 | 54.277 |
| $k$-Means++-init | 0.007 | 0.440 | 0.019 | 0.001 | 0.467 | 01.295 | 36.088 |
| $k$-Medoids | 0.007 | 0.446 | 2.768 | 0.001 | 3.222 | 04.007 | 80.410 |

## H    ADDITIONAL RESULTS

In the following pages, we provide additional results and figures augmenting the findings of the main paper. We begin with ablation studies concerning embedding and momentum resets in Sections H.1 and H.2, respectively. An illustrative example of embedded space changes induced by BRB comes next. Then, we report ARI and NMI scores (see Section D) for our main experiments with and without pretraining. This is followed by an ablation study on how important the contraction part of soft resets are in Section H.13.

### H.1    EMBEDDING RESET ABLATION

Figure 11 shows an ablation for BRB with and without *Embedding-Reset*, for $T = 20$ and $\alpha = 0.8$ (same settings as in Experiment section of the main paper). We find small but consistent drops in average clustering accuracy for GTSRB and OPTDIGITS. Results for USPS and MNIST are mostly unaffected by embedding resets. Further, we found in initial experiments (not shown here) that embedding resets affect performance more severely for higher resets ($\alpha < 0.7$). Thus, we do not use embedding resets in BRB.

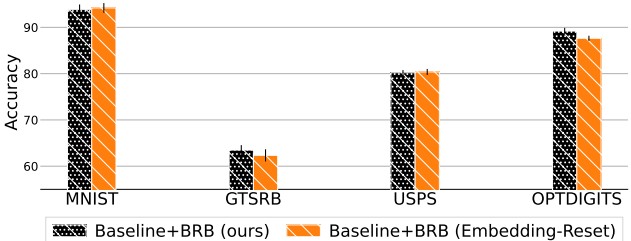

Figure 11: **Embedding reset ablation.** Clustering accuracy averaged over ten seeds for baseline methods DCN/DEC/IDEC. Using weight resets on the embedding (*Embedding-Reset*) leads to performance drops for GTSRB and OPTDIGITS, whereas performance on USPS and MNIST is unaffected by it.

## H.2 MOMENTUM RESET ABLATION

DEC and IDEC use learnable centroids that we optimize using the momentum-based Adam (Kingma & Ba, 2015) optimizer. In Figure 12, we compare BRB with and without momentum resets for $T = 20$ and $\alpha = 0.8$ (same settings as in the Experiment section of the main paper). We find that clustering accuracy drops considerably for IDEC on GTSRB and DEC on OPTDIGITS, while performance remains mostly unaffected for USPS and MNIST. To avoid large performance drops, we decided to always reset momentum in BRB for DEC and IDEC.

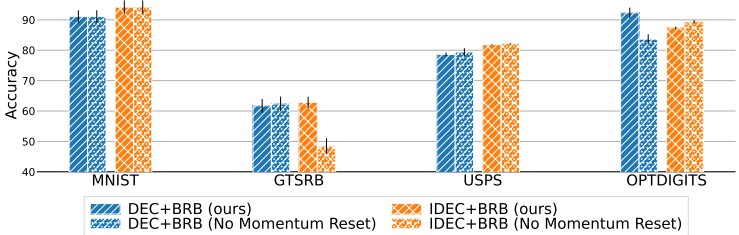

Figure 12: **Momentum reset ablation for DEC and IDEC.** Clustering accuracy averaged over 10 seeds for DEC and IDEC each. While performance for MNIST and USPS is almost unaffected if *No Momentum Reset*s are used, it drops considerably for IDEC on GTSRB and DEC on OPTDIGITS.

## H.3 RESNET BLOCK RESET ABLATION

Table 11 shows the impact of resetting the different blocks of the ResNet18 on cluster accuracy. Note that the ResNet18 consists of four ResNet blocks, where blocks with a higher number are closer to the MLP encoder used in our architecture. This means that ResNet block 4 is closest to the embedding, and ResNet block 1 is closest to the input. We see that resetting block 1 does not induce any cluster improvement over the setting without any block reset (first line). We believe this is due to the lack of sufficient change induced in the embedding. The other extreme is resetting all ResNet blocks (last line), which leads to a large drop in performance. Resetting block 4 outperforms other block reset combinations in this experiment.

Table 11: **Reset ablation for IDEC+BRB of different ResNet18 blocks on CIFAR10 with and without reset of MLP encoder and fixed** $\alpha = 0.8$ **and** $T = 10$ **for contrastive auxiliary task.** The ResNet18 consists of four ResNet blocks, where blocks with a higher number are closer to the MLP encoder, so ResNet block 4 is closest to the embedding, and ResNet block 1 is closest to the input. We report mean cluster accuracy with standard deviations over 5 runs.

| Block reset | With MLP-Reset | Without MLP-Reset |
|---|---|---|
| No Block reset | $82.39 \pm 1.29$ | $82.15 \pm 1.55$ |
| 1 | $82.36 \pm 1.12$ | $82.22 \pm 1.41$ |
| 4 | $84.56 \pm 1.69$ | $84.29 \pm 2.90$ |
| 1,2 | $82.17 \pm 0.93$ | $82.03 \pm 0.95$ |
| 4,3 | $81.29 \pm 1.82$ | $81.73 \pm 1.90$ |
| 4,3,2,1 | $68.84 \pm 5.46$ | $70.56 \pm 6.08$ |

## H.4 RECLUSTERING ALGORITHM ABLATION

To isolate the effect of different reclustering algorithms on BRB, we conduct an ablation study, evaluating $k$-Means (Lloyd, 1982), $k$-Means++ initialization (Arthur & Vassilvitskii, 2007), $k$-Medoids (Rdusseeun & Kaufman, 1987), and expectation-maximization (EM) (Dempster et al., 1977). Due to the computational cost of some algorithms, we use the smaller USPS dataset and run 10 trials for each method.

Table 12 presents the ARI for DEC, IDEC, and DCN using these reclustering methods. EM outperforms the others, achieving an average ARI of 79.46, while $k$-Means achieves 76.98. However, EM's

runtime is significantly longer, which we discuss further in Section G.2. This effect is most notable for larger datasets. Additionally, using BRB with EM is substantially more unstable, as can be seen by the increased standard errors.

Given BRB's emphasis on scalability and simplicity, we choose $k$-Means as the reclustering algorithm. Although EM may offer improved performance, its higher computational cost makes it less suitable for our application.

Table 12: ARI for DEC, IDEC, and DCN on the USPS dataset averaged over 10 seeds.

| Reclustering algorithm | DEC | IDEC | DCN | Average |
|---|---|---|---|---|
| EM | $\mathbf{79.93 \pm 2.95}$ | $82.24 \pm 2.28$ | $\mathbf{76.22 \pm 1.02}$ | $\mathbf{79.46}$ |
| $k$-Means | $75.98 \pm 1.27$ | $79.88 \pm 0.33$ | $75.09 \pm 1.54$ | $76.98$ |
| $k$-Means++-init | $73.70 \pm 1.60$ | $\mathbf{83.22 \pm 2.89}$ | $67.87 \pm 1.77$ | $74.93$ |
| $k$-Medoids | $70.27 \pm 2.51$ | $80.22 \pm 2.40$ | $75.65 \pm 1.89$ | $75.38$ |

## H.5 Voronoi Cells

To generate 2D visualizations of the embedded space, we select a subset of 4 classes on the USPS (Hull, 1994) dataset and embed these into two dimensions in Figure 13. We use DEC as clustering algorithm.

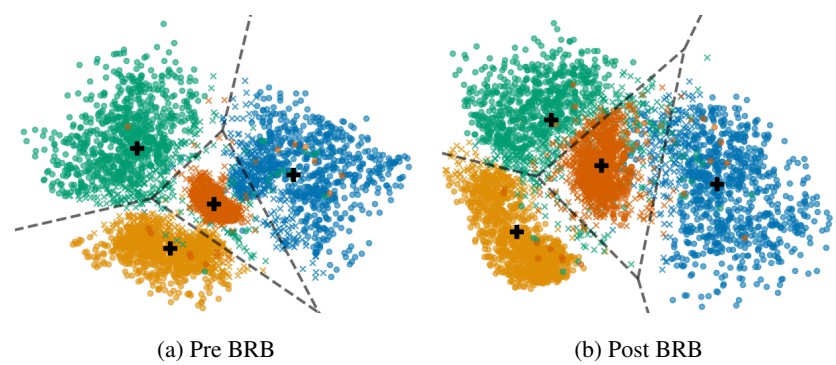

(a) Pre BRB          (b) Post BRB

Figure 13: Visualization of the Voronoi cells before and after BRB's combination of weight reset and reclustering. The plot shows how the decision boundary is moved post-BRB. Additionally, inter-class distance is increased, leading to more intermixing of clusters. Points close to the Voronoi cells are marked as "x".

## H.6 Additional Metrics with pretraining

In Figures 14a and 14b, we report additional NMI and ARI results for experiments with pretraining.

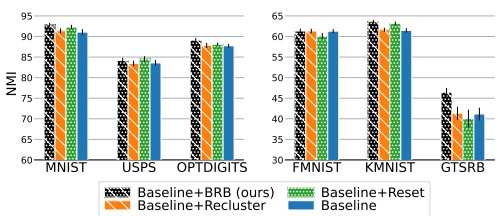 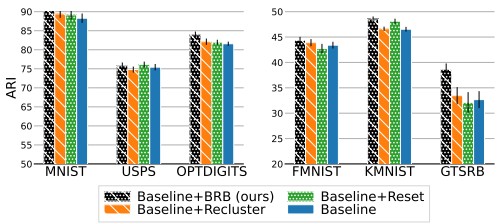

(a) **Performance experiments with pretraining measured in Normalized Mutual Information (NMI).** NMI (averaged over DEC/IDEC/DCN and ten seeds) for *BRB* and important comparison methods: *Baseline+Reset* refers to performing only weight resets (Eq. 1), *Baseline+Recluster*, refers to only reclustering and *Baseline* are DEC/IDEC/DCN without modifications.

(b) **Performance experiments with pretraining measured in Adjusted Rand Index (ARI).** ARI (averaged over DEC/IDEC/DCN and ten seeds) for *BRB* and important comparison methods as in Figure 14a.

Figure 14: Additional results for experiments with pretraining.

## H.7 Additional Metrics without pretraining

In Figures 15a and 15b, we report additional NMI and ARI results for experiments without pretraining.

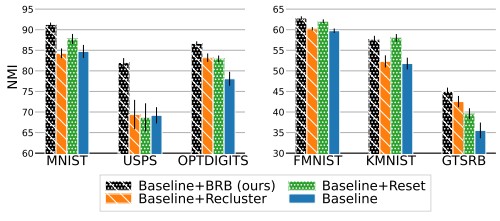 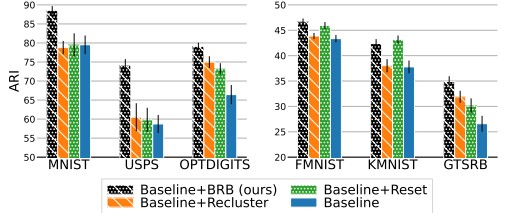

(a) **Performance experiments without pretraining measured in Normalized Mutual Information (NMI).** NMI (averaged over DEC/IDEC/DCN and ten seeds) for *BRB* and important comparison methods as in Figure 14a.

(b) **Performance experiments without pretraining measured in Adjusted Rand Index (ARI).** ARI (averaged over DEC/IDEC/DCN and ten seeds) for *BRB* and important comparison methods as in Figure 14a.

Figure 15: Additional results for experiments without pretraining.

## H.8 Detailed Ablations per Metric in Tabular Form

Tables 13, 14 and 15 show detailed results of our ablation experiments with and without pretraining.

Table 13: Clustering accuracy for various datasets.

|  | MNIST | USPS | OPTDIGITS | FMNIST | KMNIST | GTSRB | AVERAGE |
|---|---|---|---|---|---|---|---|
| DEC | 87.97±3.26 | 60.29±2.13 | 62.01±3.33 | 58.30±1.50 | 48.29±1.94 | 47.00±2.20 | 60.64 |
| DEC+BRB | 93.55±1.95 | **74.20±1.66** | **81.52±2.20** | 59.83±0.42 | 55.73±1.09 | 57.71±2.33 | **70.42** |
| DEC+Recluster | 83.41±2.37 | 48.79±4.26 | 74.97±3.16 | 57.50±2.02 | 48.69±2.00 | 53.02±2.26 | 61.06 |
| DEC+Reset | **96.12±1.46** | 53.47±5.78 | 78.64±2.18 | **59.98±0.65** | **57.73±1.08** | **60.97±2.93** | 67.82 |
| IDEC | 75.77±4.09 | 66.67±3.33 | 69.68±2.47 | 54.23±1.91 | 54.27±1.28 | 52.45±1.62 | 62.18 |
| IDEC+BRB | **85.38±2.47** | **84.50±2.68** | **80.60±2.20** | **60.49±1.70** | 56.61±1.42 | **64.10±1.25** | **71.95** |
| IDEC+Recluster | 77.71±1.78 | 74.49±2.60 | 78.12±1.89 | 53.93±1.39 | 52.92±1.57 | 62.54±1.16 | 66.62 |
| IDEC+Reset | 63.01±4.15 | 72.16±1.59 | 73.34±3.28 | 58.04±2.10 | **59.02±1.43** | 55.51±1.65 | 63.51 |
| DCN | 90.65±2.01 | 75.49±1.47 | 87.10±1.50 | 54.64±1.06 | 58.46±1.26 | 61.47±1.36 | 71.3 |
| DCN+BRB | **95.90±0.12** | **79.02±1.40** | **88.54±0.68** | **55.55±0.40** | **64.21±0.95** | **64.58±1.41** | **74.63** |
| DCN+Recluster | 92.99±1.27 | 78.82±0.56 | 88.48±0.41 | 55.51±0.78 | 63.50±1.05 | 57.94±1.55 | 72.87 |
| DCN+Reset | 86.94±2.06 | 75.58±1.29 | 84.52±1.72 | 54.29±1.07 | 63.79±1.60 | 50.40±1.16 | 69.25 |
| DEC+Pretrain | 91.56±1.98 | 78.94±1.09 | 88.21±1.08 | 60.00±0.58 | 65.28±1.04 | 62.16±2.64 | 74.36 |
| DEC+BRB+Pretrain | 91.05±2.12 | 78.63±0.60 | **92.32±1.70** | 59.64±0.78 | **65.97±1.14** | 61.73±2.24 | 74.89 |
| DEC+Recluster+Pretrain | 90.68±1.84 | 78.65±0.63 | 90.52±1.55 | 60.32±0.50 | 65.28±0.94 | 62.15±2.63 | 74.6 |
| DEC+Reset+Pretrain | **93.69±2.15** | **79.40±0.74** | 89.02±1.37 | **60.74±0.52** | 64.37±0.87 | **62.40±2.42** | **74.94** |
| IDEC+Pretrain | 93.16±2.20 | 81.59±0.28 | 86.83±0.21 | 53.59±1.01 | 65.09±1.03 | 56.63±2.16 | 72.81 |
| IDEC+BRB+Pretrain | 94.19±2.20 | **81.87±0.14** | **87.35±0.36** | **55.96±1.13** | 65.61±1.04 | **62.84±1.89** | **74.64** |
| IDEC+Recluster+Pretrain | **94.24±2.13** | 81.08±0.19 | 86.84±0.20 | 53.61±0.93 | 65.13±1.01 | 56.94±2.34 | 72.97 |
| IDEC+Reset+Pretrain | 89.69±2.37 | 81.38±0.08 | 85.99±0.53 | 53.66±1.25 | **66.05±0.97** | 60.01±2.30 | 72.8 |
| DCN+Pretrain | 89.23±2.04 | 78.29±0.97 | 86.64±0.43 | 49.91±0.60 | 63.16±0.92 | 54.68±2.33 | 70.32 |
| DCN+BRB+Pretrain | **96.40±0.12** | **80.10±1.49** | **87.92±0.13** | 52.34±0.71 | **65.19±0.92** | **65.89±1.23** | **74.64** |
| DCN+Recluster+Pretrain | 93.86±0.78 | 77.06±0.23 | 86.06±0.55 | **52.72±1.33** | 63.76±0.83 | 56.42±2.93 | 71.65 |
| DCN+Reset+Pretrain | 90.98±2.21 | 78.64±0.79 | 87.66±0.21 | 49.22±0.62 | 64.25±0.96 | 48.48±3.36 | 69.87 |

Table 14: NMI performance comparison on various datasets.

|  | MNIST | USPS | OPTDIGITS | FMNIST | KMNIST | GTSRB | AVERAGE |
|---|---|---|---|---|---|---|---|
| DEC | 86.49±1.96 | 57.97±2.17 | 70.25±2.80 | 59.32±1.05 | 43.15±2.19 | 27.01±2.94 | 57.36 |
| DEC+BRB | 91.36±1.18 | **76.90±0.76** | **87.79±1.07** | 61.68±0.50 | 52.11±1.21 | 41.50±1.96 | **68.56** |
| DEC+Recluster | 83.00±1.85 | 45.02±4.79 | 79.21±2.30 | 59.37±0.89 | 42.93±1.71 | 37.59±2.88 | 57.85 |
| DEC+Reset | **93.26±0.88** | 49.98±7.15 | 82.75±1.32 | **62.26±0.56** | **53.11±1.18** | **46.03±2.11** | 64.56 |
| IDEC | 80.23±3.99 | 70.95±2.84 | 76.89±1.88 | 59.76±1.20 | 54.46±1.24 | 33.19±1.58 | 62.58 |
| IDEC+BRB | **91.25±0.88** | **89.39±0.93** | 85.85±1.04 | 65.17±0.73 | 58.76±1.31 | **46.39±1.06** | **72.8** |
| IDEC+Recluster | 81.52±2.67 | 82.40±1.06 | 83.14±1.05 | 60.40±0.76 | 54.80±1.42 | 44.99±1.83 | 67.88 |
| IDEC+Reset | 81.17±2.16 | 78.42±1.62 | 81.43±1.75 | 63.33±0.85 | **59.74±1.22** | 37.35±1.62 | 66.91 |
| DCN | 87.48±0.79 | 78.69±0.73 | 87.08±0.55 | 60.16±0.57 | 57.72±0.65 | 46.42±1.36 | 69.59 |
| DCN+BRB | **91.13±0.19** | 79.78±0.69 | 86.36±0.36 | **61.62±0.25** | **61.81±0.40** | **47.15±0.97** | **71.31** |
| DCN+Recluster | 88.26±0.47 | **80.72±0.35** | **87.24±0.47** | 60.91±0.37 | 59.17±0.41 | 45.03±1.57 | 70.22 |
| DCN+Reset | 88.83±0.68 | 77.72±0.55 | 84.64±0.74 | 60.54±0.83 | 61.31±0.75 | 35.28±1.91 | 68.05 |
| DEC+Pretrain | 91.85±0.91 | 84.32±0.79 | 89.58±0.67 | **62.75±0.38** | 62.89±0.42 | 48.37±2.29 | 73.29 |
| DEC+BRB+Pretrain | 91.71±0.98 | 84.00±0.46 | **91.80±1.02** | 62.16±0.70 | **64.64±0.37** | 47.80±2.11 | 73.68 |
| DEC+Recluster+Pretrain | 90.97±0.87 | 84.04±0.68 | 90.64±0.81 | 62.74±0.42 | 62.99±0.31 | 48.34±2.27 | 73.29 |
| DEC+Reset+Pretrain | **93.09±0.98** | **85.38±0.65** | 90.07±0.80 | 62.72±0.48 | 63.51±0.39 | **48.39±2.21** | **73.86** |
| IDEC+Pretrain | 94.13±0.78 | 87.14±0.41 | 88.39±0.14 | 62.51±0.82 | 63.77±0.36 | 37.31±1.95 | 72.21 |
| IDEC+BRB+Pretrain | **94.88±0.67** | **88.11±0.36** | **89.29±0.26** | **63.83±0.86** | 65.12±0.52 | **44.01±1.33** | **74.21** |
| IDEC+Recluster+Pretrain | 94.36±0.80 | 87.08±0.43 | 88.40±0.12 | 62.30±0.78 | 63.80±0.39 | 37.33±1.93 | 72.21 |
| IDEC+Reset+Pretrain | 93.55±0.77 | 88.00±0.35 | 88.29±0.27 | 62.25±1.00 | **65.63±0.48** | 41.45±2.14 | 73.2 |
| DCN+Pretrain | 87.19±0.90 | 79.29±0.67 | 85.22±0.60 | 58.62±0.64 | 57.75±0.26 | 37.82±2.25 | 67.65 |
| DCN+BRB+Pretrain | **91.81±0.19** | **80.35±0.72** | **86.22±0.24** | 57.97±0.40 | **61.01±0.31** | **47.55±1.86** | **70.82** |
| DCN+Recluster+Pretrain | 88.82±0.47 | 79.23±0.60 | 84.43±0.76 | **58.95±1.01** | 58.11±0.21 | 38.40±2.72 | 67.99 |
| DCN+Reset+Pretrain | 90.20±0.74 | 80.22±0.54 | 85.87±0.35 | 55.34±0.84 | 60.44±0.34 | 30.38±4.09 | 67.08 |

Table 15: ARI performance comparison on various datasets.

| | MNIST | USPS | OPTDIGITS | FMNIST | KMNIST | GTSRB | AVERAGE |
|---|---|---|---|---|---|---|---|
| DEC | 82.77±3.68 | 47.02±2.67 | 55.76±3.34 | 43.89±1.36 | 31.57±1.85 | 18.81±2.00 | 46.64 |
| DEC+BRB | 89.77±2.24 | **67.57±1.31** | **78.72±1.92** | 47.11±0.75 | 38.61±1.32 | 31.77±1.97 | **58.92** |
| DEC+Recluster | 76.68±2.88 | 35.94±4.74 | 69.43±3.16 | 45.18±1.30 | 30.56±1.46 | 28.37±2.49 | 47.69 |
| DEC+Reset | **93.07±1.73** | 42.49±6.58 | 72.90±2.05 | **47.68±0.74** | **39.59±1.14** | **36.76±2.78** | 55.41 |
| IDEC | 70.39±5.16 | 59.18±4.15 | 62.17±3.08 | 42.95±1.59 | 38.89±1.39 | 25.76±1.87 | 49.89 |
| IDEC+BRB | **84.72±2.09** | **83.00±2.67** | **77.01±1.93** | **49.25±1.26** | 41.89±1.20 | **36.55±1.39** | **62.07** |
| IDEC+Recluster | 72.33±2.62 | 72.34±2.56 | 73.40±1.83 | 42.84±1.06 | 38.71±1.62 | 34.65±1.52 | 55.71 |
| IDEC+Reset | 60.97±4.80 | 68.11±2.11 | 69.07±3.14 | 46.79±1.38 | **43.23±0.92** | 29.83±1.79 | 53.0 |
| DCN | 85.41±1.69 | 69.97±1.03 | 81.36±1.22 | 43.21±0.79 | 42.89±1.62 | 35.20±1.01 | 59.67 |
| DCN+BRB | **91.21±0.25** | 71.89±1.40 | 81.56±0.77 | **43.66±0.32** | 46.66±0.75 | **36.44±1.66** | **61.9** |
| DCN+Recluster | 87.33±1.03 | **73.18±0.71** | **82.08±0.54** | 43.57±0.62 | 44.79±0.62 | 32.63±1.70 | 60.6 |
| DCN+Reset | 84.38±1.73 | 68.79±0.83 | 77.68±1.57 | 43.32±0.97 | **46.69±1.10** | 23.67±1.09 | 57.42 |
| DEC+Pretrain | 88.86±1.99 | 75.54±1.37 | 83.41±1.28 | 47.05±0.62 | 47.40±0.58 | **40.09±2.70** | 63.73 |
| DEC+BRB+Pretrain | 88.28±2.17 | 74.34±0.91 | **88.12±1.91** | 46.90±0.92 | **50.05±0.59** | 38.91±2.44 | 64.43 |
| DEC+Recluster+Pretrain | 87.50±1.92 | 74.63±1.20 | 85.92±1.69 | 47.41±0.60 | 47.36±0.38 | 40.07±2.69 | 63.82 |
| DEC+Reset+Pretrain | **91.12±2.24** | **77.04±0.88** | 84.47±1.60 | **47.63±0.73** | 48.63±0.38 | 39.93±2.59 | **64.8** |
| IDEC+Pretrain | 91.73±2.03 | 79.26±0.39 | 81.84±0.20 | 43.14±0.88 | 48.00±0.75 | 30.42±2.05 | 62.4 |
| IDEC+BRB+Pretrain | **93.05±1.89** | **79.89±0.38** | **82.66±0.36** | **45.40±0.99** | 48.91±0.91 | **37.69±1.93** | **64.6** |
| IDEC+Recluster+Pretrain | 92.60±2.03 | 79.05±0.43 | 81.87±0.18 | 42.88±0.81 | 48.07±0.79 | 30.69±2.05 | 62.53 |
| IDEC+Reset+Pretrain | 88.65±2.12 | 79.72±0.35 | 81.14±0.56 | 43.33±1.18 | **49.71±0.98** | 34.29±2.32 | 62.81 |
| DCN+Pretrain | 84.26±1.91 | 71.48±1.09 | 79.47±0.66 | 40.03±0.60 | 44.24±0.55 | 27.51±2.41 | 57.83 |
| DCN+BRB+Pretrain | **92.20±0.26** | **73.19±1.78** | **80.99±0.23** | 40.83±0.45 | **46.74±0.72** | **39.37±1.68** | **62.22** |
| DCN+Recluster+Pretrain | 88.18±0.91 | 70.71±0.70 | 78.53±0.88 | **41.40±1.25** | 44.45±0.44 | 29.60±2.94 | 58.81 |
| DCN+Reset+Pretrain | 87.67±1.88 | 71.52±1.11 | 80.31±0.46 | 37.24±0.63 | 46.01±0.75 | 22.04±3.27 | 57.47 |

## H.9 DETAILED RESULTS FOR CONTRASTIVE AUXILIARY TASK AND SELF-LABELING

Table 17 shows the mean cluster accuracy over ten runs for DEC, IDEC and DCN with and without BRB. We see that BRB always improves performance and leads to more than 2% increase for DCN on CIFAR10. Using self-labeling as proposed in SCAN (Gansbeke et al., 2020) we achieve even higher cluster accuracies as stated in Table 16. While DEC does not benefit as much from BRB, we find that IDEC+BRB and DCN+BRB clearly outperform SCAN by up to 6% on CIFAR100-20.

Table 16: **Ablation results with contrastive auxiliary tasks and self-labeling**. Reporting mean cluster accuracy with standard deviations over 10 runs. Results from SCAN with self-labeling for reference (SeCu did not provide average performance results).

| Methods | CIFAR10 | CIFAR100-20 |
|---|---|---|
| DEC | $85.77 \pm 2.28$ | $\mathbf{48.37 \pm 1.39}$ |
| DEC + BRB | $\mathbf{86.98 \pm 2.28}$ | $47.18 \pm 2.63$ |
| IDEC | $85.89 \pm 2.04$ | $48.85 \pm 1.70$ |
| IDEC + BRB | $\mathbf{86.50 \pm 3.24}$ | $\mathbf{50.00 \pm 2.12}$ |
| DCN | $85.25 \pm 3.11$ | $51.13 \pm 1.65$ |
| DCN + BRB | $\mathbf{89.29 \pm 0.72}$ | $\mathbf{52.32 \pm 1.95}$ |
| SCAN (Gansbeke et al., 2020) | $87.60 \pm 0.40$ | $45.90 \pm 2.70$ |

Table 17: **Ablation performance with contrastive task only.** Reporting mean cluster accuracy with standard deviations over 10 runs. Per method, the best result is set in bold, and the overall best result is underlined. BRB enables consistent improvements for DCN and IDEC on all datasets.

| Methods | CIFAR10 | CIFAR100-20 |
|---|---|---|
| DEC | $83.00 \pm 1.65$ | $46.23 \pm 1.17$ |
| DEC + BRB | $\mathbf{83.86 \pm 1.72}$ | $\mathbf{46.63 \pm 1.32}$ |
| IDEC | $82.12 \pm 1.54$ | $46.79 \pm 1.84$ |
| IDEC + BRB | $\underline{\mathbf{84.35 \pm 1.06}}$ | $\underline{\mathbf{47.92 \pm 0.87}}$ |
| DCN | $79.64 \pm 1.98$ | $44.03 \pm 1.13$ |
| DCN + BRB | $\mathbf{82.16 \pm 0.57}$ | $\mathbf{44.89 \pm 1.49}$ |

## H.10    Detailed Analysis of Inter-CD and Intra-CD behavior in Figure 6.3

The behavior of the continually increasing intra- and inter-class distances (CD) of DCN(+Reclustering) for MNIST in Figure 6.3 can be analyzed through the silhouette score Rousseeuw (1987), which is defined as the ratio between the intra- and inter-CD. The baseline models exhibit proportional increases in intra- and inter-CD, resulting in a stable silhouette score (Figure 16). This indicates that the optimizer expands the embedding space without improving cluster separation, likely due to the absence of weight decay. In contrast, BRB's soft resets help maintain small parameter and gradient norms during training (Ash & Adams, 2020). This allows BRB to effectively reduce intra-CD while increasing inter-CD, leading to enhanced cluster separation and a higher silhouette score.

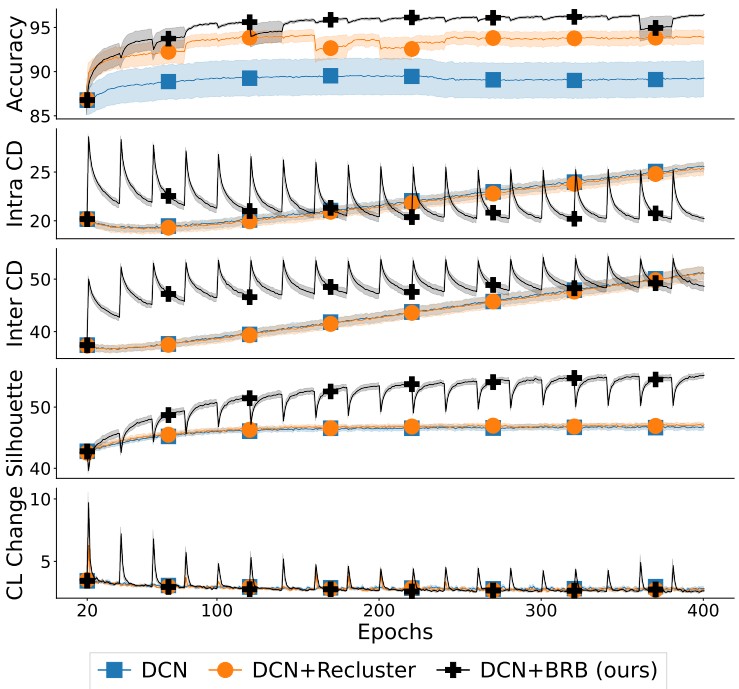

Figure 16: **Behavior of intra- and inter-CD** Figure 6.3 of our main paper augmented by showing the silhouette score in addition to average clustering accuracy, inter/intra-class distance (inter/intra-CD), and cluster label change (CL Change) for DCN on MNIST. The silhouette score is defined as the ratio between the intra- and inter-CD. The plot shows that without BRB, the proportional rise in intra- and inter-CD leads to a flat silhouette score, indicating rising embedding magnitudes without improving separation. In contrast, BRB's soft resets counteract increases in the embedding norms, allowing the cluster loss to better separate ground truth classes over time (lower intra-CD, slowly increasing silhouette score).

## H.11    Behavior of Silhouette score on predicted labels

Figures 17a and 17b show the silhouette scores for DCN on USPS and OPTDIGITS using either the ground truth or the predicted class labels. For USPS, we can observe that BRB starts improving over the baseline in terms of clustering accuracy as early as the first reset. Using the ground truth labels, this improvement also is reflected in a higher silhouette score. When using the predicted labels to calculate the silhouette score, BRB only has a substantially higher silhouette score after episode 250 despite significantly improved clustering accuracy. On GTSRB (Figure 17b), the difference between using ground truth and predicted labels is even more stark. First, note that BRB significantly outperforms the baseline in clustering accuracy. When using the silhouette score with predicted labels, the high values would suggest that both algorithms achieve strong separation, with the baseline outperforming BRB. However, the silhouette score is noticeably higher for BRB compared to the baseline when using the ground truth labels, with both the baseline and BRB achieving lower scores

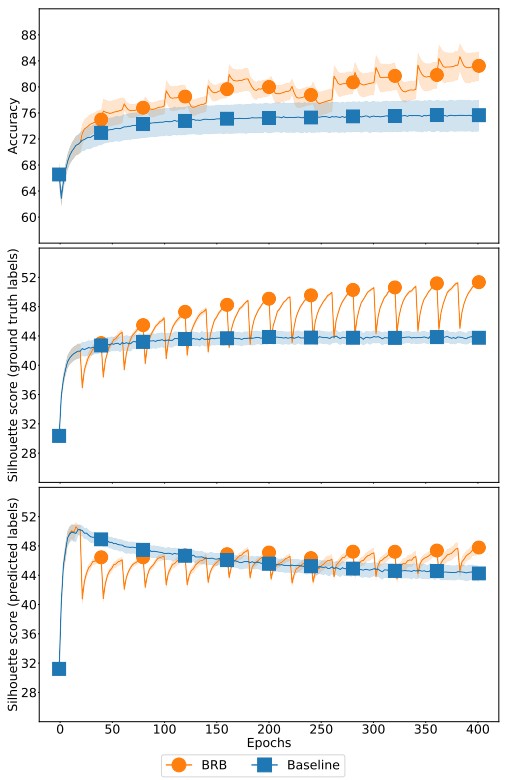 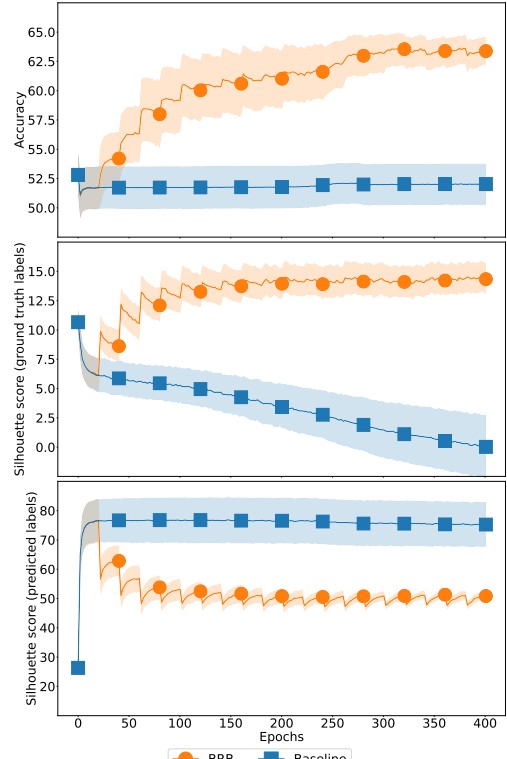

(a) **Silhouette score with predicted and ground truth labels for DCN on USPS.** The figure shows that with ground truth labels, BRB has a substantially higher silhouette score than the baseline. Using the predicted labels to calculate the silhouette score distorts the results. Results averaged over ten seeds.

(b) **Silhouette score with predicted and ground truth labels for DCN on GTSRB.** Despite significantly higher clustering accuracy for BRB, the silhouette score with predicted labels indicates superior performance for the baseline. Using the ground truth labels, BRB has a higher silhouette score than the baseline. Results averaged over ten seeds.

Figure 17: Silhouette scores with predicted and ground truth labels for DCN.

compared to using predicted labels. These experiments highlight the perils of using unsupervised metrics and predicted labels when assessing the performance of deep clustering algorithms.

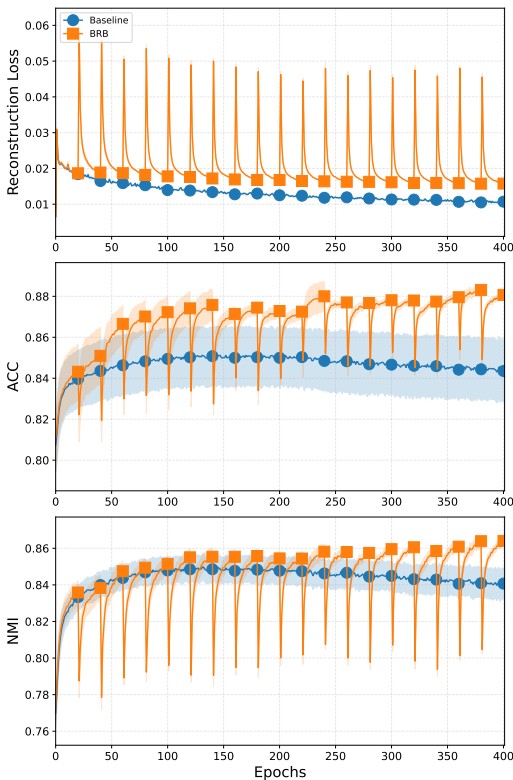

Figure 18: **Behavior of reconstruction loss for DCN on OPTDIGITS (10 seed average).** Whenever a soft parameter reset is performed, the reconstruction loss spikes before dropping down again. Overall, the loss value after a reset is on a similar level to the reconstruction loss without resets.

## H.12 ANALYSIS OF RECONSTRUCTION ERROR WITH BRB

In Figure 18, we analyze the behavior of the reconstruction loss during training of DCN on OPT-DIGITS. With each soft parameter reset, the reconstruction loss first spikes before coming down to a similar level compared to when no reconstruction loss is used. In terms of clustering accuracy, BRB's beneficial effects on the structure of the latent space outweigh the slightly higher reconstruction loss.

### H.13 ABLATION STUDY ON THE COMPONENTS OF SOFT RESETS

In Section 5, we define a soft reset according to Equation (1), which we restate below:

$$\tilde{\theta}_t^i = \iota_w(\theta_t^i) = \alpha\theta_t^i + (1-\alpha)\phi^i, \tag{19}$$

where $\theta_t^i$ are the network's parameter at step $t$ and $\phi^i$ are new parameters sampled from the initialization distribution. $\alpha$ is a hyperparameter specifying a trade-off between contracting the current parameters and the strength of the added noise. In the following, we show two ablation studies aiming to answer the following questions:

(1) How important is the contraction component of the soft reset?
(2) How important is adding noise as specified by Equation (19)?

We answer the first question by running experiments without the contraction factor for the original weights $\theta_t^i$ in Equation (19) (*Perturbation only*). To answer the second question, we add Gaussian noise with mean and standard deviation according to the current weights instead of Equation 19 (*Scaled Gaussian noise*). Figures 19a and 19b show our results for DCN trained on the datasets USPS and OPTDIGITS. We can see that removing the contraction part from Equation 19 yields substantially worse performance compared to using full soft resets, which we attribute to the ability of soft resets to balance gradients (Ash & Adams, 2020). Analyzing the gradient norms of the *Perturbation only* ablation, we indeed observe that they are substantially higher, which in turn promotes a collapse in representation and centroid norms. In contrast, the *Scaled Gaussian noise* variant of our algorithm performs similarly to soft resets on OPTDIGITS and a little worse on USPS. Thus, the results of both ablations support performing resets according to Equation (19).

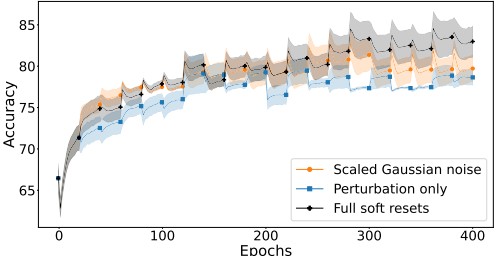

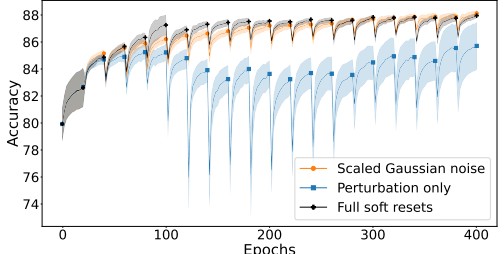

(a) **Ablation of soft reset components on USPS.** Results show the clustering accuracy for DCN, averaged over ten seeds.

(b) **Ablation of soft reset components on OPTDIGITS.** Results show the clustering accuracy for DCN, averaged over ten seeds.

Figure 19: Results of the ablation study regarding soft resets.

### H.14 ABLATION STUDY: $k$-MEANS RECLUSTERING VS. RECLUSTERING WITH DEC/IDEC ASSIGNMENTS

In addition to our experiments ablating different clustering algorithms for the *reclustering* step of BRB in Appendix H.4, we investigate whether the clustering steps of DEC and IDEC perform better than $k$-Means. DEC and IDEC produce soft assignments that are hardened by assigning a sample to the cluster with the highest assignment probability. Thus, the assignment is different from $k$-Means. Because the reclustering step of BRB is only done after the completion of an epoch and only every $T = 20$ epochs, it does not interfere with gradient updates during an epoch. As DCN already uses the $k$-Means assignment mechanism to get cluster labels, we omit it from this ablation study. Table 18 shows the performance of the vanilla DC models and their reclustering variations when training from scratch, and Table 19 shows the results with pretraining. For USPS without pretraining, the performance gap between BRB with $k$-Means and the baselines is more than 10%. For OPTDIGITS with pre-training, the BRB with $k$-Means reclustering outperforms the baselines by more than 4% for DEC. Overall, BRB with $k$-Means reclustering is either performing similar or better than the baselines. Given these results, we find that $k$-Means is indeed a sensible choice for the reclustering step of BRB.

Table 18: **Reclustering ablation without pretraining for $k$-Means vs. original DEC/IDEC reclustering**. Average clustering accuracy is computed over 10 runs using the same settings as in the main paper.

| Without Pretraining | OPTDIGITS | GTSRB | USPS |
|---|---|---|---|
| **DEC** | 60.29 | 47.00 | 62.01 |
| **DEC+BRB** w. DEC reclustering | 67.17 | **57.98** | 71.45 |
| **DEC+BRB** w. $k$-Means reclustering (ours) | **74.20** | 57.71 | **81.52** |
| **IDEC** | 69.68 | 52.45 | 66.67 |
| **IDEC+BRB** w. IDEC reclustering | 72.64 | 60.10 | 72.27 |
| **IDEC+BRB** w. $k$-Means reclustering (ours) | **80.60** | **64.10** | **84.50** |

Table 19: **Reclustering ablation with pretraining for $k$-Means vs. original DEC/IDEC reclustering**. Average clustering accuracy is computed over 10 runs using the same settings as in the main paper.

| With Pretraining | OPTDIGITS | GTSRB | USPS |
|---|---|---|---|
| **DEC** | 88.21 | 62.16 | **78.94** |
| **DEC+BRB** w. DEC reclustering | 85.72 | **62.98** | 76.96 |
| **DEC+BRB** w. $k$-Means reclustering (ours) | **92.32** | 61.73 | 78.63 |
| **IDEC** | 86.83 | 56.63 | 81.59 |
| **IDEC+BRB** w. IDEC reclustering | 86.58 | 59.36 | 80.27 |
| **IDEC+BRB** w. $k$-Means reclustering (ours) | **87.35** | **62.84** | **81.87** |

