# OpenReview forum: "Breaking the Reclustering Barrier in Centroid-based Deep Clustering"
_ICLR.cc/2025/Conference — ICLR 2025 Poster_

### Official Review · Reviewer_JvBA · 2024-10-19

**Soundness:** 4
**Presentation:** 2
**Contribution:** 3
**Rating:** 6
**Confidence:** 4

**Summary:**

The paper addresses the problem in deep clustering of early over-commitment to sub-optimal partitions. They propose a method to combat this by periodically modifying the network weights and reclustering.

**Strengths:**

The considered problem is an important one, and is reasonably well introduced the paper.

The method is intuitive.

The experimental results on CIFAR looking convincing.

There is some helpful surrounding discussion of how the method works to improve accuracy, e.g. Figures 6 and 7.

**Weaknesses:**

The method itself is intuitive but maybe a bit simple--just slightly modifying the weights and then reclustering. The weight reset comprises two parts: contracting the current values and then adding noise. It would be good to compare the reset as per Eq (1) to just adding noise to the weights, from the same or similar distribution as the weight distribution (at least the same mean and variance). This would determine whether the contraction part is doing anything.

The comparison to SOTA methods on Cifar is reasonably convincing, but there's no such comparison on MNIST, KMNIST, USPS, FMNIST, OPTDIGITS and GTSRB. Does that mean your results are substantially worse than SOTA on those datasets?

Some information from the appendices would be better placed in the main paper. The paragraph 'implementation details' is mostly just pointers to various sections in the lengthy appendices, when it could just give the details directly. Specifically, it would be good to specify (1) the initial weight distribution, (2) the frequency across epochs that you performing resetting, and (3) which image augmentations and how exactly they are used. Presumably you just create multiple augmentations for each image and enforce them all to be in the same cluster, but I wasn't sure.

**Questions:**

What is the initial weight distribution? As this is a part of the reset mechanism, it should be specified in the main paper.

For the main results, how frequently do you reset?

Which image augmentations do you how and how exactly are they put to use?

How does your method compare to SOTA on MNIST, KMNIST, USPS, FMNIST, OPTDIGITS and GTSRB?

---

> ### Author Response · Authors · 2024-11-23
> **Rebuttal by Authors**
>
> ### Ad W1) Noise comparison, contraction ablation
>
> In line with the reviewer’s suggestion, we have added ablation experiments in Appendix H.13. Leaving out the contraction part of the soft resets results in substantially worse performance. We attribute this to the role of the contraction in reducing the gradient norms, which in turn prevents representation and centroid collapse. This is supported by research in supervised learning, where soft resets have been observed to balance gradients [4]. Additionally, we perform an ablation study where we add Gaussian noise to the weights (with the same mean and variance as the current weights) instead of doing a soft reset. This shows similar performance on OPTDIGITS but performs worse on USPS.
>
> In Appendix F, we perform an ablation where we use noise in the embedded space to evaluate whether BRB’s performance gains are due to new cluster labels obtained through perturbed weights or whether the perturbation itself improves performance. As one can see in Figure 9, simply adding scaled Gaussian noise to the embedding space degrades the performance over time
>
> ### Ad W3) Missing information in the main paper
>
> Our goal with the section about implementation details was to give an overview of what relevant implementation details are and where exact details can be found. We have improved the implementation details according to the reviewer’s feedback:
>
> - We clarified that we reset momentum terms to 0 for DEC and IDEC.
> - We added the (re-)initialization distribution to the main paper.
> - We improved the item about image augmentation in the main paper with more detail. Concretely, we clarify how we use augmentations and summarize which augmentations we use for which datasets. Additionally, we added references to the Appendix tables where we list the hyperparameters of our augmentations.
>
> ### Ad Q1) Initial weight distribution
>
> We use the Pytorch default initializer to initialize convolutional and linear layers and for soft resetting, corresponding to “Kaiming uniform” [1] initialization in Pytorch. We clarified this aspect in the main paper.
>
>
> ### Ad Q2) Frequency of resets for the main results
>
> For all experiments with and without resetting (scenario 1, scenario 2) on MNIST, KMNIST, USPS, FMNIST, OPTDIGITS, and GTSRB, we use the same values of $\alpha=0.8$ and $T=20$, which we state in the bolded part in Section 6.1. Our settings for the experiments using contrastive learning on CIFAR10 and CIFAR100-20 are $\alpha=0.7$ and $T=10$, which we list in Table 6 in Appendix E.6.
>
>
> ### Ad Q3) Image augmentations
>
> We use image augmentations to enforce consistent cluster assignments among augmented and original samples, i.e., the cluster assignments should not change due to the augmentation. This requires slightly modifying the loss functions of DEC, IDEC, and DCN. The modified losses can be found in Equations 15-17) in Appendix E.
>
> We use a different set of augmentations for the grayscale and the color datasets. For all of them, we rely on torchvision. On MNIST, KMNIST, USPS, FMNIST, and OPTDIGITS, we first resize the image before applying a random affine transformation around the center. Detailed hyperparameters for these augmentations are in Table 4 of Appendix E.4. CIFAR and GTSRB are color datasets. Consequently, we use a different set of augmentations for them, which we list in Table 5 (GTSRB, Appendix E.5) and Table 6 (CIFAR, Appendix E.6), respectively. These augmentations comprise random resized crop, color jitter, random gray scaling, random flipping, and random solarization (in this order). Note that this corresponds to the SimCLR augmentations w/o Gaussian blur, which they recommend be turned off for CIFAR [2]. We have added a summary of this information to the main paper.

---

> > ### Author Response · Authors · 2024-11-23
> >
> > ### Ad W2, Q4) Comparison to SOTA on MNIST, KMNIST, USPS, FMNIST, OPTDIGITS and GTSRB
> >
> > We see the goal of our paper in establishing the reclustering barrier as an empirical phenomenon in deep clustering and using BRB to address it. For SOTA comparisons, we focused on CIFAR10 and CIFAR100-20, as these are more relevant benchmarks in image clustering.
> >
> > Our goal with the experiments on MNIST, KMNIST, USPS, FMNIST, OPTDIGITS, and GTSRB was to isolate the effects of BRB in standardized training scenarios. These include a variety of datasets and algorithms, with pre-training and without. Breaking the state-of-the-art was not our focus for these experiments. If we were to attempt to do so, it would clearly make sense to tune hyperparameters for a specific algorithm-dataset combination instead of using the same values across all experiments. Additionally, we could likely further improve performance by using CNNs instead of feedforward autoencoders, improving the pretext task (e.g., using contrastive learning instead of reconstruction), and adding the self-labeling stage from our contrastive experiments. While this is certainly possible, we do not believe it is necessary to support the arguments and claims in our paper.
> >
> > Nevertheless, BRB+DCN achieves competitive performance on MNIST with 96.40% clustering accuracy, roughly 3% below the SOTA on MNIST [3]. Algorithms at the top of the leaderboard are substantially more complex and use, for example, ensembling, adversarial training, or generative adversarial networks. Considering our simple models and training setup, we think the performance of our algorithms is reasonable.
> >
> > *Changes made to the paper according to the reviewer’s feedback are colored in blue*.
> >
> >
> >
> > [1] He, Kaiming, et al. "Delving deep into rectifiers: Surpassing human-level performance on imagenet classification." Proceedings of the IEEE international conference on computer vision. 2015.
> >
> > [2] Chen, Ting, et al. "A simple framework for contrastive learning of visual representations." International conference on machine learning. PMLR, 2020.
> >
> > [3] https://paperswithcode.com
> >
> > [4] Ash, Jordan, and Ryan P. Adams. "On warm-starting neural network training." Advances in neural information processing systems 33 (2020)

---

> > > ### Comment · Reviewer_JvBA · 2024-11-26
> > >
> > > Thank you for the reply. I think the ablation studies of adding noise help make the results more convincing. It might be helpful to reference them in the main paper if there is space.
> > >
> > > I just have a further question regarding the performance on MNIST etc. If FFNs are not optimal for deep clustering (which sounds reasonable), why did you use them in your experiments? Do you expect that the reclustering barrier would still apply and this reset method would still be effective with different architectures, and also ensembling (presumably referring to SPC, Mahon et al. 2021) and adversarial training?

---

> > > > ### Author Response · Authors · 2024-11-27
> > > >
> > > > ### Ad) Referencing noise ablation in the main paper
> > > >
> > > > We agree and rewrote Section 6.4 of our main paper to include a reference and a summary of our findings from the experiment the reviewer suggested.
> > > >
> > > > ### Ad Q1) FFN networks for deep clustering
> > > >
> > > > FFNs allow studying the properties of deep clustering algorithms in isolation of other influencing factors, clarifying the subsequent analysis. This is why they are still used in many papers introducing new deep clustering methods, e.g., [5] and [6].
> > > >
> > > > ### Ad Q2) Reclustering barrier in different architectures
> > > >
> > > > In our contrastive experiments on CIFAR10 and CIFAR100-20, we already used a convolutional architecture (ResNet), showing performance improvements for the different deep clustering algorithms when used with BRB. Similar to SPC [7], the deep clustering algorithm SeCu [8] makes use of ensembling. SeCu uses ensembling in the form of multiple clustering heads, where each head uses a different number of clusters, leading to a more diverse clustering. Our results using SeCu show that BRB can improve the performance of deep clustering algorithms that use cluster ensembles.
> > > >
> > > > |              | ACC       | NMI       | ARI       |
> > > > | ------------ | --------- | --------- | --------- |
> > > > | **SeCu**     | 88.88     | 80.31     | 78.73     |
> > > > | **SeCu+BRB** | **89.01** | **80.53** | **79.03** |
> > > >
> > > > Regarding adversarial training, we observe that adversarial clustering approaches can also suffer from rapid early gains and saturating performance (Figures 9, 11, 12 in [9]). When this type of saturation occurs at a level that is not close to optimal performance, adding BRB to the training would likely allow a centroid-based algorithm to make further gains.
> > > >
> > > >
> > > > [5] Leiber, Collin, et al. "The dipencoder: Enforcing multimodality in autoencoders." Proceedings of the 28th ACM SIGKDD Conference on Knowledge Discovery and Data Mining. 2022.
> > > >
> > > > [6] Stirn, Andrew, and David A. Knowles. "The VampPrior Mixture Model." arXiv preprint arXiv:2402.04412 (2024).
> > > >
> > > > [7] Mahon, Louis, and Thomas Lukasiewicz. "Selective pseudo-label clustering." KI 2021: Advances in Artificial Intelligence: 44th German Conference on AI. 2021.
> > > >
> > > > [8] Qian, Qi. "Stable cluster discrimination for deep clustering." Proceedings of the IEEE/CVF International Conference on Computer Vision. 2023.
> > > >
> > > > [9] Mrabah, Nairouz, Mohamed Bouguessa, and Riadh Ksantini. "Adversarial deep embedded clustering: on a better trade-off between feature randomness and feature drift." IEEE Transactions on Knowledge and Data Engineering 34.4 (2020)

---

> > > > > ### Author Response · Authors · 2024-11-29
> > > > >
> > > > > Dear Reviewer,
> > > > >
> > > > > Thank you for your valuable feedback. Your comments further improved the quality of our work and strengthened our results.
> > > > >
> > > > > We hope we have addressed your concerns. Please feel free to reach out if you have any further questions before the review period ends in 3 days. If you are satisfied with the revisions, kindly consider updating your score.
> > > > >
> > > > > Thank you again for your time and consideration.

---

> > > > > > ### Comment · Reviewer_JvBA · 2024-12-01
> > > > > >
> > > > > > Hi, what is the dataset for this table of results in your reply? Thanks.

---

> ### Author Response · Authors · 2024-12-02
>
> The dataset is CIFAR10 without using self-labeling.

---

### Official Review · Reviewer_VgYg · 2024-10-28

**Soundness:** 3
**Presentation:** 3
**Contribution:** 2
**Rating:** 5
**Confidence:** 5

**Summary:**

This paper addresses the issue of rapid early performance gains followed by quick saturation in deep clustering (DC). It introduces the concept of the "reclustering barrier" and explores when and why this phenomenon occurs. To overcome it, the authors propose the Break the Reclustering Barrier (BRB) algorithm, which avoids early over-commitment to initial clusters and allows continuous adaptation to reinitialized targets. The paper demonstrates that BRB improves performance across several clustering benchmarks.

**Strengths:**

- The paper addresses an important issue in deep clustering, performance saturation.
- The writing is smooth and easy to follow, with the primary contributions well-communicated.
- The empirical evaluation is thorough, and the BRB algorithm is shown to enhance performance across benchmarks.

**Weaknesses:**

1. While the concept of reclustering is mentioned early in the paper, the term can carry specific meanings in the context of deep clustering. Typically, reclustering refers to periodically or iteratively reassigning data points to clusters. However, since DC algorithms inherently involve iterative steps of representation learning and clustering, the authors should better contextualize their use of the term.

2. Building on the previous point, many existing DC methods already incorporate some form of iterative clustering. As such, the distinction between "DC" and "DC+Recluster" groups in the paper’s figures could be unclear. The authors should explicitly explain what differentiates these two groups and how their use of reclustering is unique. This will help readers better interpret these figures.

3. The core innovation of BRB lies in its use of weight resets, which the paper notes is inspired by a technique from reinforcement learning. Weight resetting is also known as a popular regularization technique in deep learning, where a portion of layer weights is periodically reinitialized during training. While applying this idea to deep clustering is somewhat novel, this contribution could be perceived as incremental.

4. Some choices in the implementation seem empirical. For example, the authors mention resetting only the last ResNet block and the MLP encoder, and they use hyperparameters such as LR = 0.001, alpha = 0.8, and T = 20. While these values are derived from extensive experimentation, it would strengthen the work to provide more intuitive or theoretical justification for these decisions. This would also make it easier to generalize the choice of hyperparameters (especially alpha) to other settings and DC algorithms.

5. The algorithms tested in the experiments are relatively older. The authors are encouraged to apply BRB to more recent deep clustering methods to better demonstrate its effectiveness and relevance. This would enhance the impact of the work and align it with the latest advancements in the field.

6.  The paper uses widely-known data augmentation techniques such as contrastive learning and self-labeling, which are not novel,  in the main results. Therefore, combining these techniques and comparing BRB with SOTA algorithms like SCAN, GCC, and SeCu may not represent a fully fair comparison. The authors should clarify whether the improvements stem from BRB itself or the use of these auxiliary techniques, ensuring a more balanced evaluation of the proposed method.

Minor:
Some grammar error:  (line 128) Its output are => its output is

**Questions:**

In addition to the questions raised in the Weaknesses section:

1. alpha is a critical hyper parameter. From Figure 8, it appears that α = 0.9 yields better performance across various settings. It is unclear why the authors recommend α = 0.8 instead. Additionally, should the choice of α = 0.9 or 0.8 remain consistent across all DC algorithms, or is there room for algorithm-specific tuning? Clarifying whether this parameter can or should be personalized for different algorithms would enhance understanding.

2. How does the approach extend to deep clustering methods that are not centroid-based?

---

> ### Author Response · Authors · 2024-11-23
> **Rebuttal by Authors**
>
> ### Ad W1, W2) Usage of the term “reclustering”
>
> We agree with the reviewer that DC algorithms already iteratively reassign data points over the course of the training. When we refer to “reclustering”, we refer to a complete reassignment of all points in the embedded space via some clustering algorithm, e.g., k-means.
>
> To improve clarity and reduce ambiguity, we added additional information on what we mean by “reclustering” in three places: 1) Abstract, 2) Introduction, and 4) Section 4 “The reclustering barrier”.
>
> ### Ad W3) Incremental contribution
>
> Our core contributions are three-fold. First, we identify and quantify (in terms of inter-CD and intra-CD) the reclustering barrier in centroid-based deep clustering. In our eyes, the reclustering barrier in itself is of substantial interest to the deep clustering community, as we discuss in Section 7 of our paper under “Broader insights for deep clustering research”. Second, we apply soft resets in conjunction with a k-means reclustering of the latent space and moment resets. While we agree that the mere application of resets would be an incremental contribution, we want to emphasize that naive soft resetting of the encoder’s weights does not result in performance gains, as we show in Figure 4. This indicates that BRB is a non-trivial extension of plain soft resets. Lastly, our third contribution is the empirical investigation of why BRB alleviates the reclustering barrier (cf. Figures 6 and 7 of the paper). The underlying mechanisms of BRB are highly specific to deep clustering and differ substantially from the effect of resets in reinforcement learning, where they mitigate plasticity loss [1], or supervised learning, where soft resets improve generalization when fine-tuning a pre-trained network [2].
>
> Our work also contains other contributions, such as presenting DEC, IDEC, and DCN baselines that are substantially stronger than previously used in the literature (e.g., in [3]) or the release of strong pre-trained models for other researchers to build on. In summary, we believe we make a worthwhile contribution to the current state of deep clustering algorithms.

---

> > ### Author Response · Authors · 2024-11-23
> >
> > ### Ad W4, Q1) Intuitive/theoretical justification for $\alpha$ and $T$
> >
> > The reviewer is correct in their assessment in that the choices of our hyperparameters are empirical. In particular, tuning the learning rate based on experimental results is standard practice in deep learning. While BRB does add two more hyperparameters ($\alpha$ and $T$) that must be tuned empirically, we believe this to be a small cost compared to its potential gains, especially when compared to other current SOTA algorithms. For example, the SeCu adds ten additional hyperparameters that must be tuned. Lastly, we believe that we support our choices with extensive experiments.
> >
> > From the perspective of intuition, $\alpha$ determines the level of perturbation in the latent space, where $\alpha=0$ amounts to a complete reset, discarding all previously learned knowledge, and $\alpha=1$ leaves the latent space unchanged.  To break the reclustering barrier, we want to intuitively set $\alpha$ to a value such that a reset induces perturbations in the latent space without completely destroying its structure. Therefore, we never evaluated values below 0.5 and quickly discarded experiments with $\alpha<0.7$ empirically.
> >
> > We set the frequency parameter $T=20$ to allow the optimization procedure to recover between resets. Looking at Figure 8, there is a trade-off between both parameters: One can increase the frequency of resets, but this necessitates a higher level of $\alpha$ (Figure 8, lower left corner). Conversely, if one sets $T$ to a high value, stronger resets are possible (Figure 8, rightmost column, second from the top). Our intention with Figure 8 shows that BRB is robust to a wide range of settings. In particular, its robustness to higher learning rates confirms findings from supervised learning, where reset methods have also allowed training with higher learning rates [6].
> >
> > Figure 8 shows aggregates on a MNIST. These highlight that BRB works across various settings, but care must be taken when drawing general conclusions from a single dataset. In our experiments, we found $T=20$  and $\alpha=0.8$ to be a generally strong setting, as evidenced by using it for most of the main experiments. Nevertheless, we agree that algorithm- and especially dataset-specific tuning of $T$ and $\alpha$ would definitely increase performance further. For example, DEC with $T=10$, $\alpha=0.8$ and $lr=0.005$ reaches 98.03 accuracy on MNIST.
> >
> > We have added hyperparameter ablation studies for two more datasets (USPS and OPTDIGITS) in Appendix E.3. To summarize the results; we find that both USPS and OPTDIGITS are more sensitive to lower values of $\alpha$ and $T$ while still admitting a broad range of well-performing settings for higher values of $\alpha$ and $T$. Interestingly, while our highest learning rate of 0.005 does not perform well on MNIST, it seems to help with performance on USPS and OPTDIGITS, indicating trade-offs between datasets for setting these hyperparameters. Our values used for grayscale datasets in the main paper ($T=20$, $\alpha=0.8$, learning rate 0.001) are within a broad range of well-performing hyperparameters on MNIST, OPTDIGITS, and USPS.
> >
> > ### Ad W5) Application of BRB to newer methods
> >
> > The goal of our paper was to investigate the phenomenon of the “reclustering barrier” empirically and to introduce BRB as a method that can address this phenomenon in well-established centroid-based deep clustering methods like DCN, DEC, and IDEC. As the other reviewers pointed out, our empirical findings regarding the “reclustering barrier” offer new insights for the deep clustering community and are supported thoroughly by experiments.
> >
> > Nevertheless, we show the results of adding BRB to the current state-of-the-art algorithm SeCu on CIFAR10 [1]:
> >
> >
> > |              | ACC   | NMI   | ARI   |
> > | ------------ | ----- | ----- | ----- |
> > | **SeCu**     | 88.88 | 80.31 | 78.73 |
> > | **SeCu+BRB** | 89.01 | 80.53 | 79.03 |
> >
> > Importantly, BRB also leads to (small) performance improvement when added to SeCU. This result can likely be further improved by hyperparameter tuning, which we did not do for these results.

---

> > > ### Author Response · Authors · 2024-11-23
> > >
> > > ### Ad W6) Unfair evaluation, BRB improvements
> > >
> > > To clarify whether the improvements in our reported results stem from BRB or auxiliary techniques (contrastive learning data, augmentation, self-labeling), we refer the reviewer to the table below (excerpt of Table 1 in our paper). Here, we compare DEC, IDEC, and DCN with these auxiliary techniques against the same methods with BRB on top. We can see that in terms of clustering accuracy, adding BRB always improves performance. On average, the clustering accuracy improvement is 2.34%. Thus, the results in Table 1 show that BRB’s improved performance is not solely a consequence of using contrastive learning, data augmentation, and self-labeling.
> > >
> > >
> > > | Methods               |           |  CIFAR10  |           |           | CIFAR100-20 |           |
> > > | :-------------------- | :-------: | :-------: | :-------: | :-------: | :---------: | :-------: |
> > > |                       |    ACC    |    NMI    |    ARI    |    ACC    |     NMI     |    ARI    |
> > > | Pretraining+$k$-Means |   68.97   |   63.98   |   40.13   |   37.22   |    42.25    |   14.86   |
> > > | DEC                   |   88.29   |   80.60   |   77.23   |   50.16   |    51.66    | **35.37** |
> > > | **DEC+BRB**           | **90.57** | **82.57** | **81.18** | **50.46** |  **51.72**  |   35.05   |
> > > | IDEC                  |   88.30   |   79.50   |   77.27   |   52.73   |    52.79    |   36.79   |
> > > | **IDEC+BRB**          | **90.72** | **83.26** | **81.81** | **55.43** |  **54.81**  | **38.81** |
> > > | DCN                   |   88.55   |   81.02   |   78.17   |   53.27   |    52.13    |   37.30   |
> > > | **DCN+BRB**           | **91.23** | **83.66** | **82.42** | **56.92** |  **56.76**  | **41.15** |
> > >
> > > We are glad the reviewer points out the need for a fair comparison. This is precisely why we add the mentioned auxiliary techniques (contrastive learning data, augmentation, self-labeling) to BRB for evaluation against SOTA methods like SCAN, GCC, and SeCu. All of these methods already use at least two of the advanced representation learning techniques mentioned by the reviewer. Thus, not using them for DEC, IDEC, and DCN would make the comparison unfair. That is why we report results with self-labeling for all algorithms. Concretely, the algorithms mentioned by the reviewer use the following auxiliary techniques:
> > >
> > > - SCAN, like BRB, uses SimCLR with image augmentation for contrastive pre-training of an encoder. Our self-labeling code builds on SCAN’s, with minor modifications for centroid-based clustering (see Appendix E.1). [1]
> > > - GCC uses its own graph-contrastive loss and, thus, obligatory augmentations. Their self-labeling stage is the same as SCAN’s. [4]
> > > - SeCu uses a loss inspired by contrastive learning that relies heavily on augmentations. Just like all other methods, they use the SCAN’s self-labeling. [5]
> > >
> > > Comparing DEC, IDEC, and DCN to modern methods that use several auxiliary techniques constitutes an unfair comparison, resulting in the subpar performance reported in the literature for DEC, IDEC, and DCN. We can substantially narrow the performance between these algorithms by adding the data augmentation techniques used in SCAN, GCC, and SeCu to DEC, IDEC, and DCN. Our work shows that by adding BRB on top of DEC, IDEC, and DCN, we can obtain competitive performance to SOTA algorithms and even outperform them on CIFAR100-20.
> > >
> > > ### Ad Q2) Non centroid-based deep clustering
> > >
> > > Centroid-based deep clustering methods like DEC, IDEC, and DCN are well-established and widely used across multiple domains. They form the foundation for many recent advances in deep clustering, including state-of-the-art methods like SeCu. We focus on isolating the reclustering barrier phenomenon in centroid-based deep clustering because these methods are well-established and understood.  Other clustering paradigms like density-based or hierarchical deep clustering are relatively new and operate on fundamentally different principles, including but not limited to the loss term and geometric space used. While extending our analysis to these methods is a valid and novel direction for future work, it was beyond the scope of our current investigation. Nevertheless, we agree that research on whether similar phenomena exist in non-centroid-based clustering methods is an important direction for future work.
> > >
> > > *Changes made to the paper according to the reviewer’s feedback are colored in green*.

---

> > > > ### Author Response · Authors · 2024-11-23
> > > >
> > > > [1] D'Oro, Pierluca, et al. "Sample-Efficient Reinforcement Learning by Breaking the Replay Ratio Barrier." The Eleventh International Conference on Learning Representations. 2023
> > > >
> > > > [2] Ash, Jordan, and Ryan P. Adams. "On warm-starting neural network training." Advances in neural information processing systems 33 (2020)
> > > >
> > > > [3] Van Gansbeke, Wouter, et al. "Scan: Learning to classify images without labels." European conference on computer vision. Cham: Springer International Publishing, 2020.
> > > >
> > > > [4] Zhong, Huasong, et al. "Graph contrastive clustering." Proceedings of the IEEE/CVF international conference on computer vision. 2021.
> > > >
> > > > [5] Qian, Qi. "Stable cluster discrimination for deep clustering." Proceedings of the IEEE/CVF International Conference on Computer Vision. 2023.
> > > >
> > > > [6] Zaidi, Sheheryar, et al. "When Does Re-initialization Work?." Proceedings on. PMLR, 2023.

---

> > > > > ### Author Response · Authors · 2024-11-28
> > > > >
> > > > > Dear Reviewer,
> > > > >
> > > > > Thank you for your detailed feedback, which was very helpful in improving our work.
> > > > >
> > > > > We hope we've addressed your concerns. Please let us know if you have any further questions before the review period ends in 4 days. If satisfied, please consider updating your score.

---

> ### Comment · Reviewer_VgYg · 2024-11-30
>
> Thank the authors for responding and clarifying the points raised in the review.
>
> However, I still find the reclustering process discussed in the paper unclear. Based on equation (2), it appears that during reclustering, the new centroids and corresponding assignments depend solely on the perturbed embedding, without leveraging the clustering centers and assignments derived from the deep clustering methods or the reclustering process. This approach raises concerns because most deep clustering methods, whether centroid-based or not, use distinct mechanisms to generate cluster assignments. Even in centroid-based methods, enforcing clustering assignments to be generated solely from k-means could disrupt the natural assignment process inherent to the original method. After resetting the weights, it seems more intuitive to use the reclustering mechanism specific to the deep clustering algorithm for this purpose. So, an alternative approach to BRB could involve resetting the embeddings and then running the clustering step within the original deep clustering algorithm itself, which might yield similar improvements.  It is unclear why k-means is needed for the reclustering step, and this choice warrants further discussion. Specifically, I would like to know the difference in performance between the two reclustering approaches: (1) resetting embeddings followed by reclustering with the original deep clustering algorithm, and (2) the proposed BRB method. Furthermore, the paper does not sufficiently clarify how the updated centroids and cluster assignments from reclustering are utilized in the iterative process. Deep clustering methods typically incorporate these assignments into backpropagation. Does BRB follow the same procedure, or does it only produce the final clustering output?  If there is any misunderstanding on my part, I would greatly appreciate clarification from the authors.
>
> Focusing solely on centroid-based methods could also limit the applicability of the proposed method. Non-centroid-based deep clustering methods, such as deep agglomerative clustering and deep density-based clustering, are widely used in various applications. From the authors’ response to W5, it seems that applying BRB to a newer method like SeCu shows only marginal improvement, which strengthens my concerns about the broader applicability of this method.
>
> Additionally, while the authors responded to my questions in the rebuttal, the revised manuscript does not adequately address these points. For instance: I suggested including a discussion to intuitively justify the choice of critical hyperparameters, but this has not been provided. There is still no discussion of potential extensions or relationships with non-centroid-based methods. I believe similar issues, such as the “reclustering barrier,” are likely to arise in non-centroid-based approaches as well. If these discussions are included in the revised version and I have overlooked them, I would appreciate clarification.
>
> Although I commend the authors for conducting extensive experiments and presenting their work in a nice manner, the manuscript still lacks clarity in conveying its core ideas, and its contributions remain insufficiently justified.

---

> ### Author Response · Authors · 2024-12-02
>
> ### Ad) Reclustering with  the deep clustering algorithm itself
>
> We agree that "resetting the embeddings and then running the clustering step within the original deep clustering algorithm" is the most natural thing to do, and this was actually the first thing we tried. However, we found this to lead to unstable training for DEC, IDEC, and DCN. The main reason is that the previously learned centers are no longer meaningful after resetting the embedding. However, these centers are needed in DEC, IDEC, and DCN to generate the cluster assignments. Thus, we tried to compute new centers after an embedding reset with the previously learned cluster labels. While this made the training more stable, it led to less exploration and worse cluster performance. Our ablation experiments using only resets (Figure 4, DCN+Reset) also confirm this. Only resetting the embedding and using the cluster assignment of DCN can lead to large performance drops of more than 10% on GTSRB for DCN+Reset compared to just using DCN, while our DCN+BRB improves performance by 20%.
>
> Further, note that DEC, IDEC, and DCN assume spherical, Gaussian-like clusters. They use k-means to generate their initial assignments, and they apply k-Means again after training to get their final assignments. This makes using k-means as a reclustering algorithm a natural choice for these algorithms. In Appendix H.4, we evaluate different algorithms for reclustering. The results show that k-means works well for DEC, IDEC, and DCN while being both simple and efficient.
>
>
> ### Ad) The use of updated centroids and cluster assignments from reclustering in the iterative clustering process
>
> BRB only produces the final clustering output after an epoch has ended, as outlined in the pseudocode in Algorithm 1. Therefore, BRB does not interfere with the gradient updates within one epoch. Specifically, BRB produces a new embedding and, therefore, new centroids after every $T$th epoch has ended. The new embedding and new centroids are then used for subsequent optimization steps. Both the new embedding and the new centroids implicitly affect the assignments generated in the loss of deep clustering methods, but BRB does not change the calculations for the deep clustering loss or backpropagation. Additionally, we observe in our ablation experiments that just using reclustering with k-means every $T$ epochs can improve performance in many cases, see Figure 1 or Figure 4, but BRB improves performance even further. This illustrates that reclustering helps to improve performance without destroying the learning process.
>
>
> ### Ad) Focus on centroid-based methods
>
> We agree that investigating non-centroid-based deep clustering methods is interesting, but as discussed in our limitations section, we leave this for future work. We focused on centroid-based deep clustering as it is currently the largest and most cited area of deep clustering, e.g., just DEC already has over 3500 citations, IDEC has 960, and DCN has 1130. Consequently, we believe the impact of our work is highest in this area. In contrast, the areas of deep hierarchical and deep density-based clustering methods, while definitely exciting, are much smaller in terms of usage when measuring citations. To illustrate this, we conducted a Google Scholar search for the term “deep density-based clustering”, which shows that the most cited deep density-based clustering paper, “Deep density-based image clustering” by Ren et al, 2020, has 120 citations. A search for “deep agglomerative clustering” and “deep hierarchical clustering” shows the paper “Clusternet: Deep hierarchical cluster network with rigorously rotation-invariant representation for point cloud analysis” by Cheng et al, 2019, as the highest cited paper with 202 citations. While this is not an in-depth study of the size of these subfields, we merely want to illustrate that centroid-based deep clustering methods are currently more used in research than other deep clustering approaches, making them a reasonable choice to approach first.
> BRB can likely be extended to non-centroid-based deep clustering methods by choosing a suitable algorithm for its reclustering step. The soft resetting step of BRB is a very general approach that can be applied to most neural network architectures.
>
> ### Ad) Performance gains using SeCu
>
> The limited performance gains when applying BRB to SeCu are due to saturation (SeCu is already close to supervised learning performance) and limited hyperparameter tuning. The performance can likely be further improved.

---

> > ### Author Response · Authors · 2024-12-02
> >
> > ### Ad) Changes in the manuscript
> >
> > We have added further intuitive discussion of the hyperparameters $\alpha$ and $T$ in Appendix E.3, in addition to the already existent discussion in Appendix E.2. We agree with the reviewer that a more detailed analysis of $\alpha$ and $T$ in the main paper would be ideal but want to point out that the ICLR page limit is ten pages, which we are already exhausting. We mention non-centroid-based methods in our limitations but have not included our discussion with the reviewer in our manuscript as we were unsure whether the reviewer's concerns were sufficiently addressed.
> >
> > We sincerely appreciate the high-quality discussion we had with the reviewer. Hopefully, we clarified the reviewer’s remaining concerns, particularly regarding the reclustering step. We currently cannot edit the submission but will clarify our procedure in the camera-ready version of the paper. Concretely, we will update the camera-ready version of our manuscript as follows:
> >
> > 1. Clarify the reclustering procedure in the paragraph “Reclustering”. In particular, we will mention what motivates our choice of k-means, why it may work better than using the reclustering mechanism specific to the deep clustering algorithm, and how it affects the loss and backpropagation.
> > 2. Shorten Section 6.1 to include more discussion of $\alpha$ and $T$ in the paragraph “Additional experimental results” (Section 6.2).
> > 3. Include the reviewer’s suggested experiment regarding the reclustering with the original assignment procedure in the camera-ready version of our paper.
> > 4. Expand Appendix B.1 by explaining how BRB affects the losses of DEC, IDEC, and DCN.
> > 5. Add a Section B.4 in the Appendix to include the main points of our discussion concerning the application of BRB to non-centroid-based deep clustering.

---

> ### Comment · Reviewer_VgYg · 2024-12-02
>
> I sincerely thank the authors for discussing my concerns and committing to implementing the suggested changes in the camera ready version. I still believe the "reclustering step" remains a tricky aspect that should be carefully clarified and thoroughly discussed in the revised manuscript. Regarding the comments on "new centers", I believe that leveraging the natural assignment process inherent to the original deep clustering methods can also effectively generate the new centers. Without a clear justification for the reclustering step, the novelty of this work may be undermined. Regarding the discussion of hyperparameters, I highly recommend that the authors provide more comprehensive guidelines to assist practitioners in selecting appropriate values for the proposed method.

---

> ### Author Response · Authors · 2024-12-04
>
> ### Ad reclustering step)
>
> Yes, we agree and, therefore, have already clarified our use of the term “reclustering” in the manuscript (highlighted in green). Based on our discussion with the reviewer, we will add further details to Section 4, explaining how BRB affects the loss and backpropagation steps for centroid-deep clustering algorithms. Additionally, we provide pseudocode in Algorithm 1, which shows that the reclustering step of our BRB algorithm happens outside the training loop of the deep clustering methods.
>
> Another way to think about our reclustering step is to think of it as reinitializing the clustering after the weight reset every $T$th epoch. As mentioned in our previous response, DEC, IDEC, and DCN all use k-means to initialize their cluster centers. So, reinitializing the centers with k-means after a strong embedding change is a reasonable choice.
>
> ### Ad new centers)
>
> Based on the reviewer's feedback, we conducted an experiment comparing the performance of the “natural assignment process” of DEC and IDEC with our k-means reclustering approach. We conducted 10 runs for each configuration using the same settings as in the paper. We reported the average clustering accuracy (ACC) below for three datasets (OPTDIGITS, GTSRB, and USPS), with and without pretraining. Please note that DCN already uses k-means for its assignment and centroid update step, making k-means the “natural assignment process” of DCN. This motivates our focus on DEC and IDEC.
>
> We can see that BRB with the  “natural assignment process” can indeed improve over the baseline. Still, it is outperformed by BRB with k-means reclustering (ours) in the majority of cases.  For OPTDIGITS with pre-training, the “natural assignments” are outperformed by our BRB with k-means reclustering by more than 5% for DEC. For USPS without pre-training, the performance gap between BRB with k-means and “natural assignments”  widens to 10%. Together with the fact that BRB improved performance over the baseline for 88.10 % of all experiments we conducted, we provide enough evidence that reclustering with k-means works well for DCN, DEC, and IDEC. From our perspective, this is expected, as both k-means and DCN, DEC, and IDEC assume spherical Gaussian clusters.
>
> | With Pretraining | OPTDIGITS | GTSRB | USPS |
> | --- | --- | --- | --- |
> | IDEC | 86.83 | 56.63 | 81.59 |
> | IDEC+BRB with natural assignment process | 86.58 | 59.36 | 80.27 |
> | IDEC+BRB (ours) | **87.35** | **62.84** | **81.87** |
>
> | With Pretraining | OPTDIGITS | GTSRB | USPS |
> | --- | --- | --- | --- |
> | DEC | 88.21 | 62.16 | 78.94 |
> | DEC+BRB with natural assignment process | 85.72 | **62.98** | 76.96 |
> | DEC+BRB (ours) | **92.32** | 61.73 | **78.63** |
>
> | Without Pretraining | OPTDIGITS | GTSRB | USPS |
> | --- | --- | --- | --- |
> | IDEC | 69.68 | 52.45 | 66.67 |
> | IDEC+BRB with natural assignment process | 72.64 | 60.1 | 72.27 |
> | IDEC+BRB (ours) | **80.6** | **64.1** | **84.5** |
>
> | Without Pretraining | OPTDIGITS | GTSRB | USPS |
> | --- | --- | --- | --- |
> | DEC | 60.29 | 47 | 62.01 |
> | DEC+BRB with natural assignment process | 67.17 | **57.98** | 71.45 |
> | DEC+BRB (ours) | **74.2** | 57.71 | **81.52** |

---

> > ### Author Response · Authors · 2024-12-04
> >
> > ### Ad Hyperparameters)
> >
> > We would like to highlight that based on your feedback, we have conducted additional hyperparameter studies on two more datasets. In total, we have now conducted 2880 runs to analyze the hyperparameter sensitivity of BRB with respect to the reset strength $\alpha$, reset frequency $T$, and learning rate. Please refer to Appendix sections E.2 and E.3 for results and interpretation. The hyperparameter study on three datasets clearly shows that values for reset strength $\alpha$ between 0.7 and 0.9 and reset frequency $T$ between 10 and 40 all work equally well when choosing a learning rate that is the same or slightly lower than the learning rate used during pretraining. In these experiments, the pretraining lr was 0.001. The best-performing learning rates were 0.001, and the slightly lower 0.0005.
> >
> > For all experiments with an autoencoder backbone (MNIST, FMNIST, KMNIST, USPS, OPTDIGITS,  GTSRB), we used the same learning rate and parameters for BRB with $T=20$ and $\alpha = 0.8$.
> >
> > To further confirm that these ranges for BRB generalize to completely different settings, we want to highlight that we used similar hyperparameters for CIFAR10 and CIFAR100-20 with contrastive learning, with $\alpha=0.7$ and $T=10$. Note that these are within the strong performing intervals we identified previously. $\alpha=0.7$ and $T=10$ were successfully used for the three different deep clustering algorithms: DEC, IDEC, and DCN.
> >
> > All of the above results indicate that BRB is highly stable with respect to its hyperparameters. These results can now assist practitioners in selecting appropriate values for their use case. For example, a practitioner can start by using a more conservative reset strategy, with less frequent resets $T \in \{20, 40\}$ and less strong resets $\alpha \in \{0.8, 0.9 \}$. The values of $T=20$ and $\alpha = 0.8$ we used should provide a good starting point. If the clustering results do not change at all compared to a baseline, the reset strength can be increased by specifying a lower $\alpha < 0.8$ and/or increasing the frequency of resets $T < 20$.
> >
> > A very practical way to monitor BRB’s behavior during training is to look at loss curves. The strength of the resets can be seen by monitoring the loss of the method during training and is clearly visible as bumps in the loss curve, analogous to the bumps we show for the metrics in Figure 2. If soft resets are too strong or too frequent, training will diverge, and the loss will not improve or diverge. If the soft resets are too weak or not frequent enough, the loss will not change at all. If BRB’s hyperparameters are in a reasonable range, like $\alpha$ between 0.7 and 0.9 and reset frequency $T$ between 10 and 40, then the impact on the loss curve should be clearly visible.

---

> > > ### Author Response · Authors · 2024-12-04
> > >
> > > Below, we summarize our discussion with the reviewer
> > > 1.  **Terminology Clarification - "Reclustering"**
> > >  In response to the reviewer, we disambiguated the term in 1) Abstract, 2) Introduction, and 4) Section 4 “The reclustering barrier” of our paper, and explicitly defined "reclustering" as a complete reassignment of all points in the embedded space via a clustering algorithm (e.g., k-means).
> > > 2. **Addressing Incremental Contribution Concerns**
> > >  Our three core contributions are:
> > >    - Identifying and quantifying the "reclustering barrier".
> > >    - Applying soft resets with k-means reclustering and moment resets and showing that applying soft resets naively does not work.
> > >    - Empirically investigating BRB's mechanisms.
> > > 3. **Hyperparameter Justification**
> > >  We acknowledged the empirical nature of hyperparameter selection and added explanations for our key hyperparameter settings $\alpha$ and $T$. We added additional hyperparameter sensitivity studies on USPS and OPDTIGITS. Our choices of $\alpha=0.8$ and $T=20$ provide a good starting point for practitioners.
> > > 4. **Application to Newer Methods**
> > >  We implemented BRB on top of the current state-of-the-art algorithm SeCu and showed modest performance improvements that can likely be improved further with more hyperparameter tuning.
> > > 5. **Fairness of Evaluation and BRB improvements for contrastive learning**
> > >  The reviewer referred to our usage of contrastive learning, data augmentation, and self-labeling as potentially constituting an unfair comparison. We clarified that the modern algorithms we compare against (SCAN, GCC, SeCu) use the same auxiliary techniques that we use for training BRB on CIFAR10. BRB’s improvements are not solely due to these auxiliary techniques, as we highlight in Table 1 of our paper.
> > > 6. **Extension to Non-Centroid-Based Methods**
> > >  We agree with the reviewer that extending BRB to non-centroid-based deep clustering is an exciting area for future work, which we noted in the limitations section of our paper. Our focus on centroid-based deep clustering is motivated by the prevalence of these methods in current research.
> > > 7. **Reclustering Mechanism**
> > >  The reviewer criticized our method BRB for using k-means for reclustering and pointed out that we should recluster using the original deep clustering algorithm's assignment process. We explained our conceptual reasoning for choosing k-means, namely that DCN, DEC, and IDEC learn a k-means-friendly embedded space (spherical Gaussian clusters) and use k-means to generate their initial assignments. Additionally, we found that using k-means instead of the original deep clustering algorithm's assignment process improved the exploration of clustering solutions. We provided additional experiments supporting our findings and arguments.
> > > 8. **Commitment to Future Improvements**
> > >  Crucial elements of our discussion with the reviewer fell outside the time window where the manuscript could be changed. Therefore, we proposed specific changes for the camera-ready version of the manuscript based on our discussion with the reviewer.

---

### Official Review · Reviewer_XSEo · 2024-11-04

**Soundness:** 3
**Presentation:** 3
**Contribution:** 2
**Rating:** 6
**Confidence:** 4

**Summary:**

The paper discusses 'reclustering barrier', a common and important phenomenon that occurs in centroid-based deep clustering methods, where performance quickly saturates in the very early stage of the training phase. The authors carefully discuss when and why reclustering barrier happens and propose BRB which prevents this early commitment to initial clustering and gain performance over the training with the reinitializing clusters after every few epochs. They apply BRB on top of several existing deep clustering methods to verify its advantage. The paper contains simulation results to support its main claims.

**Strengths:**

### SIGNIFICANCE:
This work is important because it seeks to overcome an important phenomenon commonly seen in centroid based deep clustering methods and this can be studied further over other recent deep clustering methods (also not centroid based as they may also face the similar issue).

### ORIGINALITY:
This work is novel because it demonstrates the application of BRB over some existing methods that improves the performance further.

### CLARITY:
This work is well-written and well-organized. It clearly explains the key components of the method and their functions. For example, the discussion about the hyperparameter $T$, $\alpha$ and learning rate is complete and important for the reproducibility and understanding of this work. In addition, I also have some questions regarding this method (see the next section).

### QUALITY:
The quality of this work is good. The work includes carefully designed simulations that are sufficient to support the claim of the paper.

**Weaknesses:**

1. The biggest weakness is the baselines (on top of which BRB has been applied) are too old to compare, the latest one is from 2017. It would be good to see applying BRB on top of one or two recent DC method (like SCAN, GCC or SeCu mentioned in the paper) and report those results. This would provide a clearer comparison to the state-of-the-art.

2. Hyperparameter sensitivity. It would be good to report similar hyperparameters (T, α and learning rate) sensitivity (figure 8) to other one or two datasets besides MNIST.

**Questions:**

1. You are computing inter-CD and intra-CD to measure variation within ground truth classes (for example, in figure 2). Why is this so important? What about this metrics for the predicted clustering by BRB? I suggest to include a comparison of these metrics for both ground truth classes and predicted clusters, which would provide a more complete picture of BRB's performance.

2. Again for figure 2, it seems even after 400 epochs the clustering labels are changing, but the accuracy is not. What is happening here? Are the clustering labels flipping only?

3. Is BRB optimizing based on the clustering metric (say accuracy)? Or is it updating for a specific epochs (say 500) and after that you are computing the relevant metrics? In other words, is BRB using trues labels while training? At what points the evaluation metrics are computed?

4. Accuracy, NMI or ARI, all of these metrics are supervised and need true labels. However, in practice, we don't have any true labels to compute these metrics. Can you include some unsupervised metric (say Silhouette Coefficient) in your evaluations and comparisons, in addition to the supervised metrics? This would provide insight into the method's performance in more realistic unsupervised scenarios.

5. As BRB modifying the encoding space after each reclustering step, the encoding space may degrade and may not represent the original data point anymore, did you verify it with the original data point with reconstruction loss? I suggest to report reconstruction loss for one or two datasets comparing to baselines, which would help to understand the underlying encoding structure.

6. Momentum resets section is not clear enough, can you elaborate more like how are you exactly re-initializing the momentum for the cluster centers after reclustering?

---

> ### Author Response · Authors · 2024-11-23
> **Rebuttal by Authors**
>
> ### Ad W1) Applying BRB on top of more recent DC methods
>
> The goal of our paper was to investigate the phenomenon of the “reclustering barrier” empirically and to introduce BRB as a method that can address this phenomenon in well-established centroid-based deep clustering methods like DCN, DEC, and IDEC. As the other reviewers pointed out, our empirical findings regarding the “reclustering barrier” offer new insights for the deep clustering community and are supported thoroughly by experiments.
>
> Nevertheless, we show the results of adding BRB to the current state-of-the-art algorithm SeCu on CIFAR10 [1]:
>
> |              | ACC   | NMI   | ARI   |
> | ------------ | ----- | ----- | ----- |
> | **SeCu**     | 88.88 | 80.31 | 78.73 |
> | **SeCu+BRB** | **89.01** |  **80.53** |  **79.03** |
>
> Importantly, BRB also leads to (small) performance improvement when added to SeCU. This result can likely be further improved by hyperparameter tuning, which we did not do for these results.
>
> ### Ad W2) Hyperparameter sensitivity on another dataset
>
> We have added hyperparameter ablation studies for two more datasets (USPS and OPTDIGITS) in Appendix E.3. Our experiments show that both USPS and OPTDIGITS are more sensitive to lower values of $\alpha$ and $T$ while still performing well for settings with higher values of $\alpha$ and $T$. Interestingly, while our highest learning rate of 0.005 does not perform well on MNIST, it seems to help with performance on USPS and OPTDIGITS, indicating trade-offs between datasets for setting these hyperparameters. Our values used for grayscale datasets in the main paper ($T=20$, $\alpha=0.8$, learning rate 0.001) are within a broad range of well-performing hyperparameters on MNIST, OPTDIGITS, and USPS.
>
> ### Ad Q1) Importance of inter-CD and intra-CD
>
> This is an excellent question. To see why using the ground truth classes for calculating the inter-CD and intra-CD is crucial, note that it is possible to place clusters arbitrarily far apart in latent space. Optimizing the clustering objective will always yield compressed and well-separated clusters, particularly for algorithms that do not use an auxiliary reconstruction loss like DEC. Therefore, these algorithms will always produce good unsupervised scores, even if we have a poor clustering result in terms of ground truth labels. The high flexibility of deep clustering algorithms regarding the embedding makes the global structure within the embedding and, therefore, unsupervised scores less meaningful.
>
> Figure 3 can help to build an intuition. If we were to measure inter-CD and intra-CD with the predicted cluster labels, we would get a better score before applying BRB (Stage 1, “Pre-BRB”) than after BRB (Stage 3, “Post-BRB”) due to superior cluster separation. However, using inter-CD and intra-CD with predicted labels does not account for wrongly assigned samples. BRB facilitates the re-assignment of these potentially mislabeled samples by temporarily changing the structure in the latent space, followed by reclustering. This effect can only be measured by calculating the inter-CD and intra-CD with ground truth labels.
>
> ### Ad Q2) Label change without clustering accuracy change
>
> We compute label assignments using the Hungarian method before computing the cluster label change. Therefore, purely flipping labels does not cause the behavior observed in Figure 2. Instead, the resets flip labels between certain wrongly assigned points back and forth. Due to their position in the embedded space, the reset is not strong enough to facilitate assignment to the correct cluster. Note that this is an artifact of the GTSRB dataset. For example, Figure 6 shows that cluster label changes reduce over time for MNIST. Figure 6 also shows another case (OPTDIGITS) where the cluster label changes induced by the resets affect accuracy, but the algorithm converges to a similar accuracy value. Thus, the above findings indicate that the clustering algorithm has converged to a robust solution to the perturbations caused by the soft resets.
>
> ### Ad Q3) Use of true labels and calculation of metrics
>
> BRB does not use the true labels. We use the true labels for calculating relevant metrics, e.g., clustering accuracy, which we compute after each epoch.

---

> > ### Author Response · Authors · 2024-11-23
> >
> > ### Ad Q4) Unsupervised metrics
> >
> > In Appendix H.11, we added additional experiments tracking the silhouette score using predicted labels on USPS and GTSRB. On both datasets, BRB significantly outperforms the baseline in terms of clustering accuracy. When using the silhouette score with predicted labels, the difference in performance only becomes apparent after episode 250, whereas BRB's improved performance is clearer when using the ground truth labels. On GTSRB, using the predicted labels in the silhouette score indicates strong cluster separation, with the baseline substantially outperforming BRB. When using the ground truth labels, however, the silhouette score shows noticeably better performance for BRB, mirroring the improvement in clustering accuracy. This shows the perils of evaluating deep clustering algorithms with unsupervised metrics in a controlled benchmark setting, supporting our argument in "Ad 1) Importance of inter-CD and intra-CD".
> >
> > ### Ad Q5) Reconstruction loss
> >
> > We have added additional experiments showing the reconstruction error for DCN on OPTDIGITS in Appendix H.12. The experiments show that with each soft parameter reset, the reconstruction loss spikes before dropping down to a value close to the baseline.
> >
> > ### Ad Q6) Clearer explanation of momentum resets.
> >
> > For DEC and IDEC, we reset the Adam momentum terms to zero after the reclustering step of BRB. The motivation, therefore, is that both algorithms parameterize the centroids and optimize them with gradient descent. Note that the soft reset does not directly change the centroids because we do not apply it to the centroid parameters. In contrast, the subsequent k-means reclustering does change the centroids. Consequently, the Adam moments accumulated up to this epoch may not point in the direction of the steepest descent for the changed centroid locations. For DCN, this issue is irrelevant because its centroids are not updated with gradient descent. We clarified our description of the moment resets in the paper by stating that we reset the moments to zero.
> >
> > *Changes made to the paper according to the reviewer’s feedback are colored in orange*.
> >
> > [1] Qian, Qi. "Stable cluster discrimination for deep clustering." Proceedings of the IEEE/CVF International Conference on Computer Vision. 2023.

---

> > > ### Author Response · Authors · 2024-11-28
> > >
> > > Dear Reviewer,
> > >
> > > We sincerely appreciate your feedback and time. Your thorough review raised excellent questions, for which we provided extensive answers.
> > >
> > > We are reaching out to inquire if you have any further questions we could clarify within the discussion period (which ends in 4 days). We hope we've addressed your concerns. If so, please consider updating your score.

---

> ### Comment · Reviewer_XSEo · 2024-11-30
>
> Dear Authors,
>
> Thank you for addressing my concerns in your responses. However, I still have some concerns that I would like to address:
>
> 1. Regarding W2, It seems there is still room for searching the best hyperparameters across each dataset (which may end up with better results).
> 2. Regarding Q1, then how one can calculate inter-CD and intra-CD where the ground truth labels are not available (consider a pure unsupervised setting)? The same goes for supervised metric like accuracy, nmi etc. I believe there should be always an unsupervised metric (along with reconstruction loss so that the embedding points (and clusters) don't be meaningless) to evaluate clustering and that also should be with the predicted labels, not with the ground truth labels.
> 3. Regarding Q4, I don't quite get figure 17. If the accuracy goes higher, that means predicted labels are matching with the ground truth labels, right? So why, BRB outperforms baseline in respect to SC while considering ground truth but not for the predicted ones (even there are big difference specially for 17b)?
> 4. Regarding Q5, it seems reconstruction loss of BRB is worse than the baselines, that is another concern of the embedding points to be less meaningful.

---

> ### Author Response · Authors · 2024-12-02
>
> ### Ad 1) Tuning hyperparameters for each dataset
>
> We agree with the reviewer that “searching the best hyperparameters” could further improve the performance of BRB. Nevertheless, we want to emphasize that BRB’s improved performance over the various baselines in different training settings despite not tuning its hyperparameters is a clear strength and important for its practical application: As our hyperparameters work well across multiple datasets with different characteristics, practitioners can assume that they also lead to good results for the purely unsupervised setting mentioned by the reviewer.
>
> ### Ad 2) Unsupervised metrics
>
> In a pure unsupervised setting, one would likely have to rely on using predicted labels as there are no better alternatives. As we are evaluating benchmark scenarios, we have ground truth labels available and thus use widely accepted metrics (NMI, ARI, ACC) within the deep clustering literature for fairly comparing algorithms. The evaluation with ground truth labels is currently the standard procedure in deep clustering. NMI, ARI, and ACC are used to benchmark performance in papers of recent algorithms ((see [1, 2, 3]) and are recommended in up-to-date surveys ([4, 5, 6]).  As per the reviewer’s request, we did include experiments showing the reconstruction loss and Silhouette score with predicted labels. We also discussed the dangers of using predicted labels (see Ad 3) of this comment and Ad Q1) in our rebuttal). The scenario outlined by the reviewer is an unsupervised model selection problem, where we must choose the best model in a purely unsupervised setting. Problems of this kind are an orthogonal area of research and require, e.g., specifically developed frameworks for hyperparameter tuning, such as [7].
>
> ### Ad 3) Figure 17 “So why, BRB outperforms baseline in respect to SC while considering ground truth but not for the predicted ones”
>
> The reviewer is right that higher accuracy implies a better match with ground truth labels. When using the predicted labels, this fact is irrelevant. A high Silhouette score with predicted labels for the baseline means that the learned clusters are compact and well-separated. This does not imply that they are a good fit for the ground truth labels, as the lower accuracy for the baseline shows. Thus, the baseline finds compact clusters but merges the wrong ground truth classes together. For BRB, the resets artificially keep the clusters more spread out in latent space (see Intra-CD and Inter-CD in Figure 2). This naturally leads to a worse Silhouette score with predicted labels due to less compact clusters but enables reassignments late in training, which allows for a better clustering given the ground truth.
>
> Figure 3 of our paper illustrates potential problems with unsupervised metrics in latent space. “Stage 1, Pre-BRB” roughly corresponds to how the baseline clusters look like for the baseline in Appendix H.11: Well-separated given the predicted labels, but without consideration for ground truth labels. For the Silhouette score, this can lead to high values when using the predicted labels but low values when using the ground truth labels. “Stage 3, Post-BRB” corresponds to how the clustering looks like for BRB: Less compact and separated clusters. This decreases the Silhouette score using the predicted labels but allows for the reassignment of points close to the borders of the Voronoi cells late in training, improving clustering accuracy.
>
> ### Ad 2, 4) Worse reconstruction loss, meaningless embedding points (and clusters)
>
> The meaningfulness of an embedding is tied to a particular task, which, in our case, is clustering. Reconstruction can, but does not necessarily, indicate a meaningful embedding for clustering. BRB has slightly worse reconstruction loss than the baselines, making its embedding less meaningful for reconstruction tasks. This does not imply that BRB’s embedding has less meaning with respect to clustering tasks. Indeed, the improved clustering metrics such as ACC, NMI, and ARI indicate that BRB’s embedding is actually more meaningful for clustering. To further illustrate this point, consider the deep clustering algorithm DEC, which is one of the algorithms we use in our experiments. DEC does not use any reconstruction loss during the clustering phase but can still learn meaningful embeddings that correspond to the ground truth labels despite a growing reconstruction loss. We currently cannot edit the submission, but for the camera-ready version of our paper, we will include reconstruction loss plots for DEC to illustrate this point.

---

> > ### Author Response · Authors · 2024-12-02
> >
> > [1] Van Gansbeke, Wouter, et al. "Scan: Learning to classify images without labels." European conference on computer vision. Cham: Springer International Publishing, 2020.
> >
> > [2] Zhong, Huasong, et al. "Graph contrastive clustering." Proceedings of the IEEE/CVF international conference on computer vision. 2021.
> >
> > [3] Qian, Qi. "Stable cluster discrimination for deep clustering." Proceedings of the IEEE/CVF International Conference on Computer Vision. 2023.
> >
> > [4] Zhou, Sheng, et al. "A comprehensive survey on deep clustering: Taxonomy, challenges, and future directions." ACM Computing Surveys 57.3 (2024)
> >
> > [5] Lu, Yiding, et al. "A survey on deep clustering: from the prior perspective." Vicinagearth 1.1 (2024)
> >
> > [6] Huang, Huajuan, et al. "Deep image clustering: A survey." Neurocomputing 599 (2024)
> >
> > [7] Fan, Xinjie, et al. "On hyperparameter tuning in general clustering problems." International conference on machine learning. PMLR, 2020.

---

> ### Comment · Reviewer_XSEo · 2024-12-03
>
> Dear Authors,
>
> Thank you for further addressing my concerns. I do not have any other concern to discuss. It would be good to see discussion on reconstruction loss in the revision.

---

> ### Author Response · Authors · 2024-12-03
>
> Dear Reviewer XSEo,
>
> thank you for your response. We are glad that we could further clarify your concerns.
>
> To address your last remaining concern regarding the relationship of reconstruction loss and clustering performance (meaningfulness of an embedding), we have conducted the following experiment:
>
> We compare the performance in terms of clustering accuracy (ACC) and reconstruction loss (REC-Loss) of the deep clustering methods DEC and IDEC averaged over 10 runs below. Note, that DEC and IDEC are almost identical algorithms, except that IDEC keeps the reconstruction loss after pretraining. The comparison between DEC and IDEC thus shows the importance of the  reconstruction loss in relation to the clustering accuracy.
>
> If the reconstruction loss is the only thing that determines how meaningful an embedding is, then DEC should not be able to capture any meaningful cluster information and perform much worse than IDEC. However, from the tables below we see that while the REC-Loss of DEC is between 10 to 1000 times larger than the REC-Loss of IDEC it still performs well overall. For the datasets OPTDIGITS, GTSRB and FMNIST DEC clearly outperforms IDEC, even though the REC-Loss is several orders of magnitude higher.
>
> | **DEC** | MNIST | OPTDIGITS | USPS | KMNIST | GTSRB | FMNIST |
> | --- | --- | --- | --- | --- | --- | --- |
> | REC-Loss | 46.08 | 1.19 | 1.66 | 63.29 | 1.08 | 134.78 |
> | ACC | 91.56 | **88.21** | 78.94 | 65.28 | **62.16** | **60** |
>
> | **IDEC** | MNIST | OPTDIGITS | USPS | KMNIST | GTSRB | FMNIST |
> | --- | --- | --- | --- | --- | --- | --- |
> | REC-Loss | 0.1 | 0.01 | 0.02 | 0.25 | 0.18 | 0.08 |
> | ACC | **93.2** | 86.83 | **81.6** | 65.09 | 56.63 | 53.59 |
>
> We will include this discussion on the importance of the reconstruction loss into the camera ready version of our paper, and hope this sufficiently alleviates your last remaining concern regarding our work. If so, please consider increasing your score.

---

### Official Review · Reviewer_xoWL · 2024-11-06

**Soundness:** 3
**Presentation:** 4
**Contribution:** 4
**Rating:** 8
**Confidence:** 4

**Summary:**

The authors explore a (known) problem when training a clustering model from scratch as follows. The architecture has an encoder followed by a clustering head. With end-to-end optimization, the encoder is updated by the gradients which flow back to it through the clustering head. This means that if and when the clustering head is saturated at a local minima, the encoder will stop learning. This can happen before the encoder has learnt a good encoding, resulting in a model which ceases to learn further despite performing poorly.

The authors demonstrate the problem, their method (a periodic weight perturbation strategy with cluster centroid reinitialization), show experiments comparing against relevant baselines, and analyze the behavior of their method.

**Strengths:**

The problem is well presented and salient to the field. As far as I am aware, the proposed method is novel. The analysis is thorough, with ablations as well.

**Weaknesses:**

It would be better if the work studied some harder datasets/tasks, such as ImageNet-1k or some taxonomic image datasets (iNaturalist, BIOSCAN, etc.). A lot of the datasets considered are simpler digit classification datasets. The work could be further improved by evaluating on other modalities.

**Minor points**

Some references are badly formatted and need some attention to correct them.
- Incorrect casing ("em algorithm", "hungarian method", "Spagcn", "rna")
- DOIs shouldn't include the domain name and should actually be links (the latter is fixed by importing the latex `doi` package in main.tex) "DOI: https://doi.org/10.24432/C50P49" -> "doi:[10.24432/C50P49](https://doi.org/10.24432/C50P49)". When the DOI is a link, you don't need to also include the URL field.
- It's good for readers if you can include links in the references (ideally as DOIs) as much as possible, so the reader can easily navigate to the article being cited (and know they are looking at the work you meant to cite)

**Questions:**

Why do the authors only consider the CIFAR100 supersets (CIFAR100-20) instead of clustering 100 classes?

---

> ### Author Response · Authors · 2024-11-23
> **Rebuttal by Authors**
>
> We are thrilled to hear the reviewer’s positive feedback regarding our work! In particular, we are glad about the reviewer’s assessment regarding our analysis and ablations.
>
> ### Ad) Minor weaknesses
>
> We thank the reviewer for thoroughly reading our paper. The typos they pointed out are now corrected. Additionally, we fixed the DOIs in the updated version of our paper.
>
> ### Ad) Harder datasets, CIFAR100 clustering
>
> The focus of our evaluation is broadly two-fold. First, we evaluate various simpler datasets (MNIST, KMNIST, USPS, FMNIST, OPTDIGITS, and GTSRB) to understand the effects of BRB under different conditions in a unified benchmarking environment. Second, we show that the combination of BRB and DEC, IDEC, and DCN can achieve surprisingly strong performance on well-established benchmarks for image clustering.
>
> We use CIFAR10 and CIFAR100-20 because these are the most commonly used image clustering benchmarks. They have moderate computational requirements and allow us to evaluate BRB against a range of relevant competitors. Clearly, using the more realistic real-world datasets such as iNaturalist and BIOSCAN is an exciting direction for future work. However, evaluating our method on them is beyond the scope of our work due to a lack of well-established benchmarks for clustering algorithms on these datasets. Lastly, we want to point out that our baseline methods have been used for different types of real-world data already, as we point out in our introduction. Some of these include challenging datasets, such as noisy financial data [1] or medical data with missing values [2].
>
> [1] Choi, Stephen, and Tyler Renelle. "Deep learning price momentum in US equities." 2019 International Joint Conference on Neural Networks (IJCNN)
>
> [2] Kalweit, Maria, et al. "Patient groups in Rheumatoid arthritis identified by deep learning respond differently to biologic or targeted synthetic DMARDs." PLOS Computational Biology 19.6 (2023)

---

### Author Response · Authors · 2024-11-23
**General response**

We thank the reviewers for their time, thoughtful comments, and valuable suggestions for improving the paper. We appreciate the positive comments regarding the presentation of our manuscript. Their comments highlight the novelty and thorough motivation behind our work. We are pleased that the reviewers recognize the comprehensiveness of our experimental analysis, showing the effectiveness of our method BRB in addressing the challenge of early cluster performance saturation.

We respond to each reviewer in detail below. In the following, we briefly summarize the improvements to our work:

- We clarified several aspects of our method by adding more detail on how we recluster, how we perform momentum and weight resets, and how we use augmentations. We also disambiguated the term reclustering more thoroughly.
- We added additional hyperparameter sensitivity experiments for two datasets, USPS and OPTDIGITS.
- We included experiments using the predicted labels for the silhouette score on USPS and GTSRB.
- We added a figure in the Appendix showing the reconstruction loss behavior when performing resets on OPTDIGITS.
- We performed an additional ablation on the importance of the contraction of our soft resets on OPTDIGITS and USPS.
- We implemented BRB on top of a more modern algorithm, showing performance improvements.

Our manuscript has been updated accordingly.

---

### Comment · Area_Chair_m76f · 2024-11-25
**The author-reviewer discussion period is ending soon**

Dear reviewers,

If you haven’t done so already, please engage in the discussion as soon as possible. Specifically, please acknowledge that you have thoroughly reviewed the authors' rebuttal and indicate whether your concerns have been adequately addressed. Your input during this critical phase is essential—not only for the authors but also for your fellow reviewers and the Area Chair—to ensure a fair evaluation.
Best wishes,
AC

---

### Author Response · Authors · 2024-12-04
**Second General response**

We thank all the reviewers for their contributions and valuable feedback on our work. Their thoughtful questions resulted in a highly relevant discussion and improved our paper. We believe we satisfied most of the reviewers’ concerns through extensive additional experiments and further clarification. Below is a summary of our takeaways from the discussion phase:

- Reviewer XSEo mainly had questions regarding our evaluation metrics and hyperparameters. We included the desired metrics and supported our choice of using clustering accuracy (ACC), Normalized Mutual Information (NMI), and Adjusted Rand Index (ARI) both with literature and empirically. We added hyperparameter sensitivity analyses on two more datasets. The reviewer indicated that our answers fully addressed their concerns.
- Reviewer VgYg raised concerns about the reclustering step of BRB, our hyperparameter choices, BRB with SOTA algorithms, and our evaluation setup. In reply, we clarified the term reclustering in our manuscript. We provided extensive details on the reclustering step and additional experiments that further justified our design choices for BRB. Moreover, we added new hyperparameter sensitivity experiments and clarified the intuition behind these hyperparameters. Lastly, we addressed the concerns regarding our evaluation setup and provided results for combining BRB with a SOTA algorithm. With our experiments and analysis of the “natural assignment process” vs. k-means cluster assignments, we have addressed the reviewer’s remaining major concern.
- Reviewer JvBA was concerned with the form of our weight reset, SOTA results, and the clarity of our presentation. We conducted an ablation study according to the reviewer’s specifications, supporting the use of soft resets, and added additional information to our main paper. Furthermore, we discussed our experiment setups and implementation specifics in more detail and added experiments with BRB and a SOTA deep clustering algorithm. According to the reviewer, we answered all of their questions.
- Finally, we answered the questions of reviewer xoWL.

---

### Meta-Review · Area_Chair_m76f · 2024-12-21

**Metareview:**

This paper examines a pathological issue in certain centroid-based deep clustering methods, referred to as a reclustering barrier, and proposes a straightforward yet effective solution. While the analysis is not rigorous, the paper offers an intuitive explanation for the reclustering barrier, and the suggested remedy consistently enhances the baselines across various benchmark datasets. Some concerns were raised regarding the clarity of the term "reclustering"; however, the authors’ rebuttal appears to address this to some extent. I strongly encourage the authors to incorporate the additional comments from the reviewers in the final version of the manuscript.

**Additional Comments On Reviewer Discussion:**

While there were some initial concerns regarding the details or presentation, most were addressed during the rebuttal. One reviewer remained negative throughout the discussion, and while I acknowledge the validity of the criticism to some extent, I do not believe it outweighs the reasons to accept the paper.

---

### Decision · Program_Chairs · 2025-01-22

Accept (Poster)